# Selective attention and sensitivity to auditory disturbances in a virtually real classroom

**Orel Levy[1], Shirley Libman Hackmon[1], Yair Zvilichovsky[1], Adi Korisky[1], Aurelie Bidet-Caulet[2], Julie B Schweitzer[3,4], Elana Zion Golumbic[1]\***

[1]The Gonda Brain Research Center, Bar Ilan University, Ramat Gan, Israel; [2]Aix Marseille Univ, Inserm, INS, Inst Neurosci Syst, Marseille, France; [3]Department of Psychiatry and Behavioral Sciences, University of California, Davis, Sacramento, United States; [4]MIND Institute, University of California, Davis, Sacramento, United States

## eLife Assessment

This **important** study investigates how AD(H)D affects attention using neural and physiological measures in a Virtual Reality (VR) environment. **Solid** evidence is provided that individuals diagnosed with AD(H)D differ from control participants in both the encoding of the target sound and the encoding of acoustic interference. The VR paradigm here can potentially bridge lab experiments and real-life experiments. The study is of potential interests to researchers who are interested in auditory cognition, education, and ADHD.

**\*For correspondence:**
elana.zion-golumbic@biu.ac.il

**Competing interest:** The authors declare that no competing interests exist.

**Abstract** Many people, and particularly individuals with attention deficit (hyperactivity) disorder (AD(H)D), find it difficult to maintain attention during classroom learning. However, traditional paradigms used to evaluate attention do not capture the complexity and dynamic nature of real-life classrooms. Using a novel virtual reality platform, coupled with measurement of neural activity, eye-gaze, and skin conductance, here we studied the neurophysiological manifestations of attention and distractibility, under realistic learning conditions. Individuals with AD(H)D exhibited higher neural responses to irrelevant sounds and reduced speech tracking of the teacher, relative to controls. Additional neurophysiological measures, such the power of alpha-oscillations and frequency of gaze-shifts away from the teacher, contributed to explaining variance in self-reported AD(H)D symptoms across the sample. These ecologically valid findings provide critical insight into the neurophysiological mechanisms underlying individual differences in the capacity for sustained attention and the proneness to distraction and mind-wandering, experienced in real-life situations.

## Introduction

Many people find it difficult to maintain attention to a frontal classroom lecture. Doing so requires listening, processing, and comprehending the teacher's speech over a prolonged period of time, while avoiding distraction from both irrelevant background sounds and internal mind-wandering (*Smallwood et al., 2007*; *Szpunar et al., 2013*; *Thomson et al., 2015*; *Esterman and Rothlein, 2019*). This task is supposedly even more difficult for individuals diagnosed with attention-deficit disorder or attention-deficit and hyperactivity disorder (referred to jointly as AD(H)D), which is often characterized by increase proneness to distraction and mind-wandering and poorer sustained attention (*Avisar and Shalev, 2011*; *Berger and Cassuto, 2014*; *Merrill et al., 2022*). Unsurprisingly perhaps, individuals

are most likely to seek clinical help as a result of attentional difficulties experienced in school contexts (*Litner, 2003*), and there is a tight link between attention performance and learning outcomes (*Loe and Feldman, 2007*; *Gray et al., 2017*).

However, current clinical and scientific tools used for evaluating and quantifying the constructs of 'distractibility' and 'inattention' are greatly removed from the real-life experience in organic classrooms and other ecological contexts. Commonly used approaches either rely on subjective self-report (e.g., rating scales), or are based on highly artificial computerized tests, such as Continuous Performance Tests (CPT). Unfortunately, in recent years, there is a growing understanding that these tools lack sufficient sensitivity and specificity to reliably capture the type and severity of real-life attention challenges that people face outside the lab or the clinic (*Rabiner et al., 2010*; *Narad et al., 2015*; *Hall et al., 2016*; *Murray et al., 2018*; *Mulraney et al., 2022*; *Hobbiss and Lavie, 2024*).

In attempt to create laboratory-based tasks that are more ecologically valid, and better capture the type of attentional challenges faced in real-life, recent years have seen increased use of virtual reality (VR) in cognitive research (*Parsons, 2015*; *Seesjärvi et al., 2022*). In particular, VR classrooms have been used to test different aspects of cognitive performance and distraction by ecologically valid stimuli. For example, it has been suggested that CPT tasks administered in VR classroom environments show greater sensitivity for AD(H)D classification relative to standard evaluations and can provide more refined measures of attention and distractibility (*Rizzo et al., 2006*; *Neguţ et al., 2017*; *Coleman et al., 2019*; *Parsons et al., 2019*; *Stokes et al., 2022*). However, to date, most VR classroom studies still use relatively non-ecological tasks, and rely primarily on indirect behavioral outcomes to assess aspects of attentional performance. To improve the utility of VR classrooms for understanding attention and distraction in real-life classroom learning, these platforms need to use genuine academic tasks and stimuli, to more faithfully simulate the classroom experience. Moreover, there is need to incorporate more objective metrics related to attention, such as measurements of neural activity and physiological responses, in order to obtain a more comprehensive picture of how listeners actually respond to stimuli and process the content of a lesson in a VR classroom. Here, we address this challenge by introducing a highly realistic multi-modal VR classroom platform, that simulates participating in a frontal lesson by an avatar-teacher with occasional disturbances from different types of ecological sound-events (non-verbal human sounds such as coughs and throat clearings; and artificial sounds such as phone ringtones) (see also *Levy et al., 2025*). Our novel platform also integrates measurements of neurophysiological responses, including neural activity (using electroencephalography; EEG), eye-gaze, and arousal levels, as reflected in changes in skin conductance (SC). These multi-level recordings allow us to investigate which neurophysiological metrics distinguish between students with and without a diagnosis of AD(H)D when they engage in realistic classroom learning, and which metrics reliably predict the severity of AD(H)D symptoms.

We focus on several specific neural and physiological metrics, that have been associated with aspects of attention, distraction, and information processing in more traditional attention research, and test their generalizability to this ecological context. These include: (1) Neural speech tracking of the teacher's lesson, which captures aspects of its sensory and linguistic processing, and is known to be reduced under conditions of inattention or distraction (*Ding and Simon, 2012a*; *Zion Golumbic et al., 2013*; *Huang and Elhilali, 2020*; *Holtze et al., 2021*; *Kaufman and Zion-Golumbic, 2023*). Accordingly, this metric might be expected to be reduced in individuals with AD(H)D and/or in conditions that contain external disturbances; (2) Neural event-related potentials (ERPs), transient changes in SC, and overt gaze-shifts following unexpected sound-events in the background of the classroom. These metrics are thought to reflect exogenous capture of attention, increases in arousal and potentially distraction by salient irrelevant stimuli (*Posner, 1980*; *Bidet-Caulet et al., 2015*; *Masson and Bidet-Caulet, 2019*); (3) The frequency of gaze-shifts away from the teacher, and time spent looking around the classroom, metrics associated with attention-shifts and distractibility (*Grosbras et al., 2005*; *Schomaker et al., 2017*), and are often heightened in individuals with AD(H)D (*Mauriello et al., 2022*; *Stokes et al., 2022*; *Selaskowski et al., 2023*); (4) The power of alpha- and beta-oscillations, which are often associated with increased mind-wandering or boredom (*Clarke et al., 2001*; *Boudewyn and Carter, 2018*; *Jin et al., 2019*), and some have suggested may be altered in individuals with AD(H)D (although use of spectral signatures as biomarkers for AD(H)D is highly controversial; *Gloss et al., 2016*); (5) Continuous levels of arousal, as measured by SC, which some propose are either

heightened or reduced in individuals with AD(H)D relative to their control peers (*Sergeant, 2000*; *Bellato et al., 2020*).

This novel ecological approach and the rich repertoire of neurophysiological metrics measured here, afford unique insights into the mechanistic underpinnings of paying attention in class and the disruptive effects of background sounds, in both the neurotypical and AD(H)D population.

## Results

The current dataset is extremely rich, consisting of many different behavioral, neural, and physiological responses. In reporting these results, we have separated between metrics that are associated with **paying attention to the teacher** (behavioral performance, neural tracking of the teacher's speech, and looking at the teacher), those capturing **responses to the irrelevant sound-events** (ERPs and event-related changes in SC and gaze); as well as more global neurophysiological measures that may be associated with the listeners' overall **'state' of attention or arousal** (alpha- and beta-power and tonic SC).

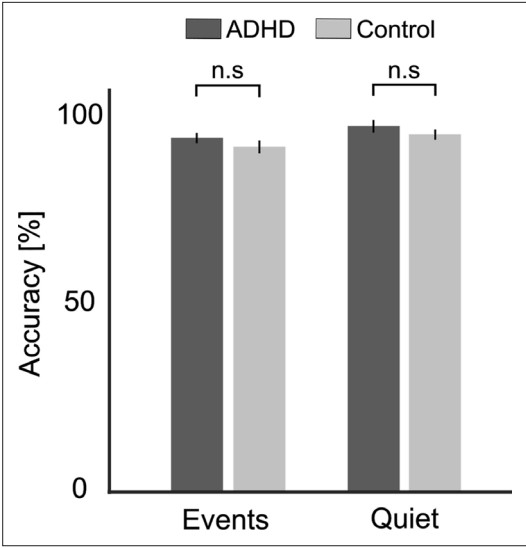

**Figure 1.** Accuracy on comprehension questions. Shown separately for the attention deficit (hyperactivity) disorder (AD(H)D) and Control groups, in the Quiet and Events conditions. Bar graphs represent average accuracy levels across participants and error bars represent the SEM. No significant differences were found between groups in either condition (n.s).

## Paying attention to the teacher

First, we tested whether there are differences between individuals with and without AD(H)D in metrics associated with focusing ones' attention on the teacher – accuracy on answering comprehension questions, neural speech tracking of the teacher's speech, and the gaze-patterns of focusing overt attention toward the teacher.

Due to the difference in number of trials statistical testing of between-group differences were performed separately for the Events and Quiet conditions (when applicable), and comparisons between conditions are evaluated only qualitatively.

### Behavioral accuracy

Participants demonstrated overall good performance on the comprehension task, achieving an average accuracy of 87.87% (±6.77% SEM). This serves as verification that participants followed the instructions, listened to teacher's speech and understood the content of the mini-lessons. Performance levels were similarly good in both the Events and Quiet conditions, indicating that the presence of occasional sound-events did not disrupt overall understanding of the mini-lessons. No significant differences in performance were found between groups, in either condition [Events: $t(47) = 1.052$, p = 0.297, Cohen's $d = 0.30$, Bayes Factor ($BF_{10}$) = 0.448 (weak support of H0); Quiet: $t(47) = 1.004$, p = 0.320, Cohen's $d = 0.28$, $BF_{10} = 0.430$ (weak support for H0); *Figure 1*].

### Speech tracking of teacher's speech

We conducted speech-tracking analysis of the neural response to the teacher's speech, using both an encoding and decoding approach. As noted in the methods section, speech-tracking analysis could only be performed reliably in the Events condition, and we used a multivariate approach to account for variance in the neural activity due to the presence of event-sounds (see *Figure 2—figure supplement 1* for comparison with univariate analysis).

The temporal response functions (TRFs) estimated in response to the teachers' speech were similar across both groups showing the typical negative peak at ~100 ms (N1), followed by a positive peak ~200 ms (P2), and the model's predictive power following the traditional mid-central topography associated with auditory responses (*Figure 2A, B*).

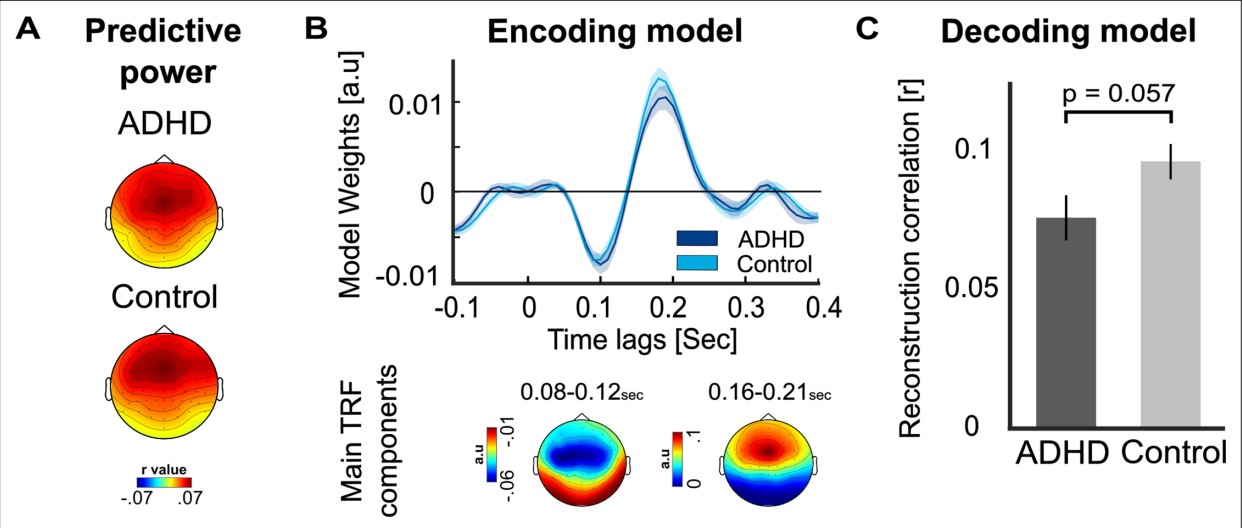

**Figure 2.** Speech tracking of the teacher in the presence of sound-events. (**A**) Topographical distribution of the predictive power values, estimated using the multivariate encoding model, in the ADHD and Control groups. No significant differences were observed between groups. (**B**) Temporal response functions (TRFs) estimated for each group in the Events condition, based on the average of electrodes FC1, FC2, and FCz, which exhibited the strongest activity for the two main components. Shaded areas represent the SEM. Below the TRFs, topographies of the main component are presented. (**C**) Bar graphs showing the average reconstruction correlation of the decoding model (Pearson's $r$) in each group. The ADHD group showed slightly lower reconstruction correlations for the teacher's speech, though the effect did not reach statistical significance (p = 0.057). Error bars represent the SEM.

The online version of this article includes the following figure supplement(s) for figure 2:

**Figure supplement 1.** Comparison between univariate and multivariate temporal response function (TRF) model.

No significant between-group differences were found in the predictive power of the encoding model (averaged across electrodes), and Bayes analysis indicates moderate support for the null hypothesis [$t(47) = 0.23$, p = 0.814, Cohen's $d = 0.067$, $BF_{10} = 0.15$]. However, when using a decoding model, which considers all electrodes in one model, we did find slightly lower reconstruction correlations for the teacher's speech in the AD(H)D group relative to controls [$t(47) = -1.948$, p = 0.057, Cohen's $d = -0.557$, $BF_{10} = 1.308$ (weak/moderate support for H1); *Figure 2C*].

## Looking at the teacher

Analysis of gaze-patterns showed that, as expected, participants spent most of the time looking at the teacher (60.2 ± 20.9% SEM of each trial). However, we also note the large variability across participants, with some focusing almost exclusively on the teacher (near 100%) and others spending less than 40% of the trial looking at the teacher. When not looking at the teacher, the next most popular places to look at were the blackboards behind the teacher, to the right and to the left (*Figure 3A*). We tested whether the amount of time spent looking at the teacher and the number of gaze-shifts away from the teacher were modulated by Group (AD(H)D vs. Control) separately in each Condition (Quiet and Events). No significant differences were found, however BF analysis indicated that the null hypotheses was not strongly supported [*Percent gaze-time at teacher:* Events Condition: $t(47) = -0.899$, p = 0.372, Cohen's $d = -0.257$, $BF_{10} = 0.397$ (weak support for H0); Quiet Condition: $t(47) = -0.597$, p = 0.553, Cohen's $d = -0.170$, $BF_{10} = 0.33$ (weak support for H0); *Number of gaze-shifts away from teacher*: Events Condition: $t(47) = 1.265$, p = 0.211, Cohen's $d = 0.361$, $BF_{10} = 0.547$ (weak support for H0); Quiet Condition: $t(47) = 0.644$, p = 0.522, Cohen's $d = 0.184$, $BF_{10} = 0.338$ (weak support for H0); *Figure 3C*].

## Responses to irrelevant sound-events

Next, we tested whether neurophysiological responses to the event-sounds themselves differed between groups. The three event-related metrics tested here were: neural ERPs, transient changes

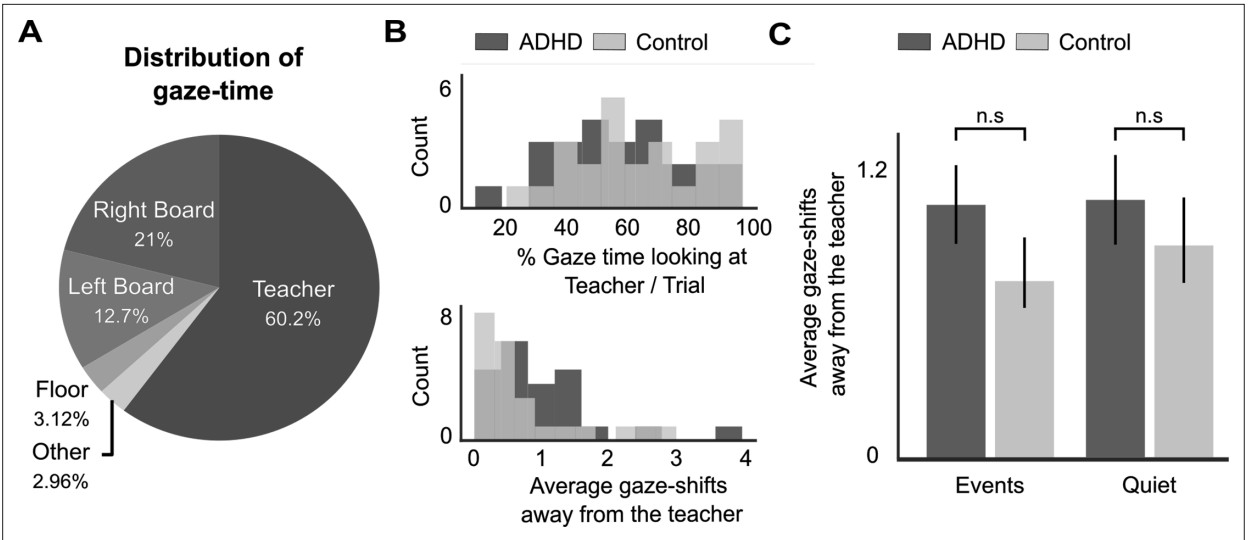

**Figure 3.** Eye-gaze results. (**A**) Pie chart representing the average amount of time that participants spent looking at different areas within the virtual reality (VR) classroom. (**B**) Distribution of the proportion of gaze-time toward the teacher (top) and number of gaze-shifts performed away from the teacher (bottom), for all participants in the attention deficit (hyperactivity) disorder (AD(H)D) and Control groups. (**C**) Bar graph represents the average number of gaze-shifts performed away from the teacher, shown separately for the AD(H)D and Control groups, and for the Quiet and Events conditions. No significant differences were found in any comparison (n.s).

in SC, and gaze-shifts following event-sounds. In addition, we compared the response to the two different event-types presented here (Artificial vs. Non-verbal human sounds).

## Neural ERPs

Visual inspection of the neural ERPs showed three prominent centro-parietal components – a negative peak around 100 ms, which corresponds to the early sensory N1; followed by two positive peaks around 240 and 350 ms, which likely corresponds to the P2 and P3 responses, respectively (*Figure 4*).

Statistical analysis comparing ERPs in the AD(H)D vs. Control groups (collapsed across both Event-types) showed that the early N1 response was significantly larger in the AD(H)D group (p < 0.05, cluster correction; *Figure 4A, B*). However, no significant differences between groups were found for the later P2 response. When comparing ERPs to Artificial vs. Non-verbal human sounds we found that both the N1 and the P2 were significantly modulated by Event-type, with a larger N1 for the Artificial sounds and a larger P2 for the non-verbal human sounds (both p < 0.002, cluster correction; *Figure 4C, D*). A mixed ANOVA performed on the amplitudes of each component confirmed these main effects, but did not reveal any significant interaction between Group and Event-type [N1: *Group* [$F(1, 47) = 6.15$, p = 0.044, $\eta^2 = 0.11$], *Event-type* [$F(1, 47) = 63.82$, p < $10^{-9}$, $\eta^2 = 0.57$], *interaction* [$F(1, 47) = 0.389$, p = 0.24, $\eta^2 = 0.008$]; P2: *Group* [$F(1, 47) = 0.84$, p = 0.36, $\eta^2 = 0.017$], *Event-type* [$F(1, 47) = 71.75$, p < $10^{-9}$, $\eta^2 = 0.6$], *interaction* [$F(1, 47) = 0.07$, p = 0.78, $\eta^2 = 0.001$]; P3: *Group* [$F(1, 47) = 0.028$, p = 0.876, $\eta^2 = 0.0005$], *Event-type* [$F(1, 47) = 0.184$, p = 0.669, $\eta^2 = 0.003$], *interaction* [$F(1, 47) = 0.166$, p = 0.684, $\eta^2 = 0.003$]; *Figure 4*].

## Event-related increase in SC

Sound-events elicited clear event-related changes in the phasic SC response, which peaked 2–3 s after the sounds, and took another ~1 s to return to baseline (*Figure 5A, B*). This response is consistent with the well-documented orienting reflex, elicited following salient events.

A mixed ANOVA showed a significant main effects of Event-type, with a higher response following artificial sounds relative to non-verbal human sounds [$F(1, 47) = 7.88$, p = 0.007, $\eta^2 = 0.143$], however there was no significant main effect of Group [$F(1, 47) = 0.4$, p = 0.53, $\eta^2 = 0.008$] or interaction between them [$F(1, 47) = 0.02$, p = 0.87, $\eta^2 = 0.0005$; *Figure 5B, C*].

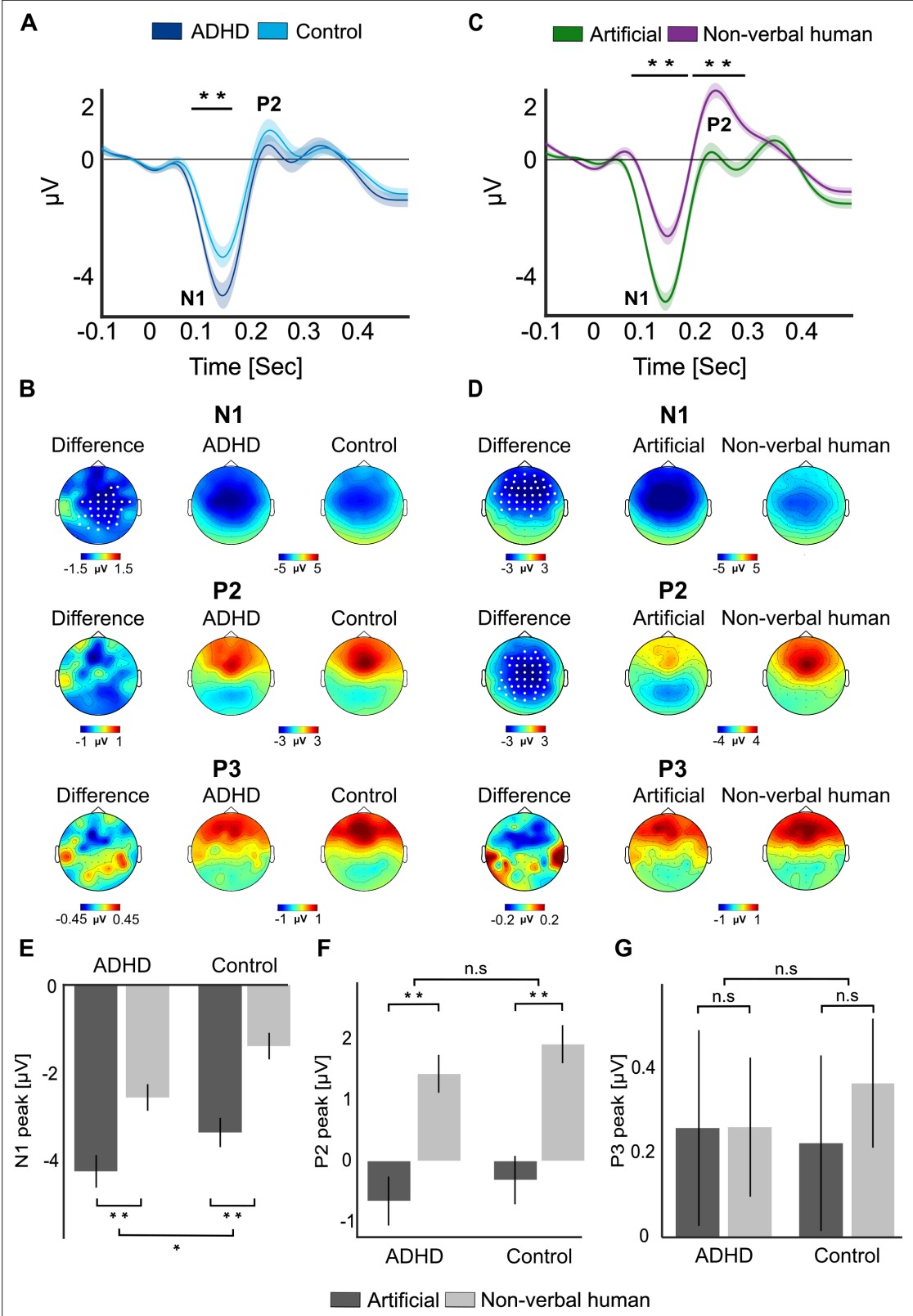

**Figure 4.** Event-related potentials (ERPs) in response to sound-events. (**A**) Grand-average ERP to sound-events, shown separately for the attention deficit (hyperactivity) disorder (AD(H)D) versus Control groups. ERPs are averaged across the cluster of electrodes where significant differences were found (see panel B). The black horizontal line indicates the time-window where significant differences between groups were found (75–155 ms). Shaded areas around the waveforms represent the SEM. (**B**) Scalp topographies of the N1 and P2 responses in the AD(H)D and Control groups, and the

*Figure 4 continued on next page*

*Figure 4 continued*

difference between them. Electrodes where significant differences between groups were found are marked in white (p < 0.05, cluster corrected). (**C**) Grand-average ERP to Artificial and Non-verbal human event-sound. ERPs are averaged across the cluster of electrodes where significant differences were found (see panel D). The black horizontal lines indicate the time-windows where significant differences between ERP to the two types of sound-events were found (67–178 and 187–291 ms). Shaded areas around waveforms represent the SEM. (**D**) Scalp topographies of the N1 and P2 responses to Artificial and Non-verbal human sounds and the difference between them. Electrode where significant differences were found are marked in white (p < 0.05, cluster corrected). (**E, F, G**) Box plots depicting the average N1, P2 and P3 responses, separately for each group (AD(H)D vs. Control) and Event-type (Artificial vs. Non-verbal human). **p < 0.001; *p < 0.05.

## Event-related gaze-shifts

We tested whether the sound-events triggered overt gaze-shifts away from the teacher, by quantifying the percentage of event-sound that were followed by at least one gaze-shift, and comparing this to control epochs taken from the Quiet condition. However, no significant differences were found, suggesting that the sound-events did not elicit more frequent overt gaze-shifts away from the teacher, relative to what might be observed when no sound-events are present [$t(48)$ = 1.34, p = 0.18; *Figure 6A*]. We also tested whether the likelihood of performing event-related gaze-shifts was different for the two Event-types (Artificial vs. Non-verbal human sounds) or in the two Groups (AD(H)D vs. Control) but did not find any significant main effects or interactions [*Group*: $F(47)$ = 0.18, p = 0.67, $\eta^2$ = 0.003; *Condition*: $F(47)$ = 2.25, p = 0.14, $\eta^2$ = 0.045; *Interaction*: $F(47)$ = 0.51, p = 0.47, $\eta^2$ = 0.01; *Figure 6B*].

## Neurophysiological metrics associated with overall attention and arousal

Besides analyzing neurophysiological metrics that can be directly associated with processing the teacher's speech or with response to the sound-events, several additional neurophysiological measures have been associated more broadly with the listeners' state of attention or arousal. These including spectral properties of the EEG (and specifically power in the alpha and beta range), the overall level of tonic and phasic SC, and spontaneous gaze-patterns.

We tested whether these metrics differed between the two groups, separately in the Quiet and Events conditions.

## Spectral EEG features

Spectral analysis of the EEG focused on the two frequency bands for which clear peaks were detected in the periodic power-spectrum, after applying the FOOOF algorithm: alpha- (8–12 Hz) and

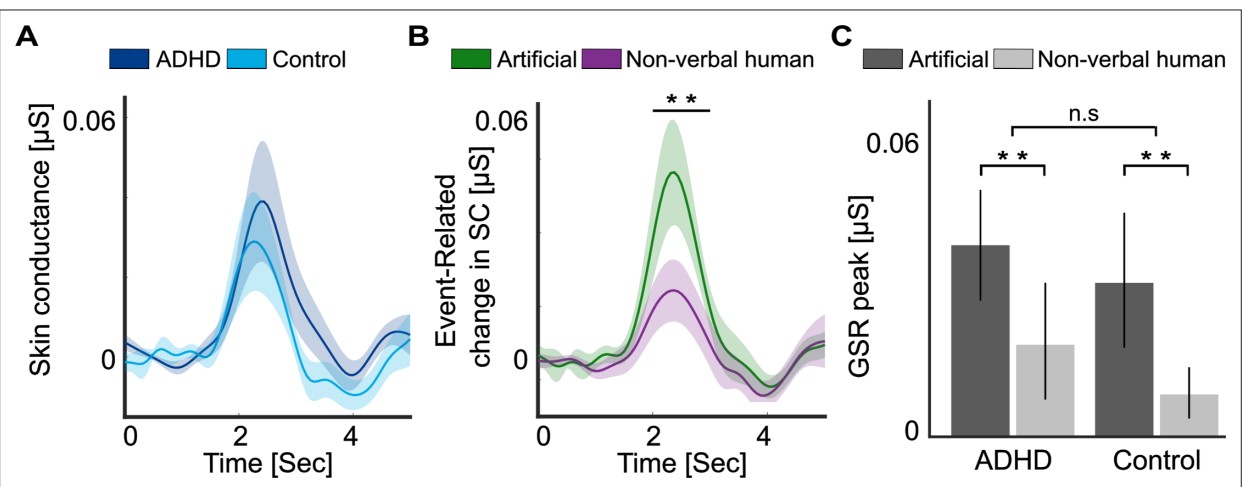

**Figure 5.** Event-related changes in skin conductance (SC). Time-course of event-related changes in phasic SC following the sound-events, shown separately for each group (**A**) and for the two sound-events types (**B**). Shaded areas around waveforms represent the SEM. Horizontal line represents the time-windows where significant differences were found. (**C**) Average levels of event-related changes in phasic SC (peak between 2 and 3 s) shown for each group and for Artificial vs. Non-verbal human sound-events. Error bars represent the SEM. **p < 0.01.

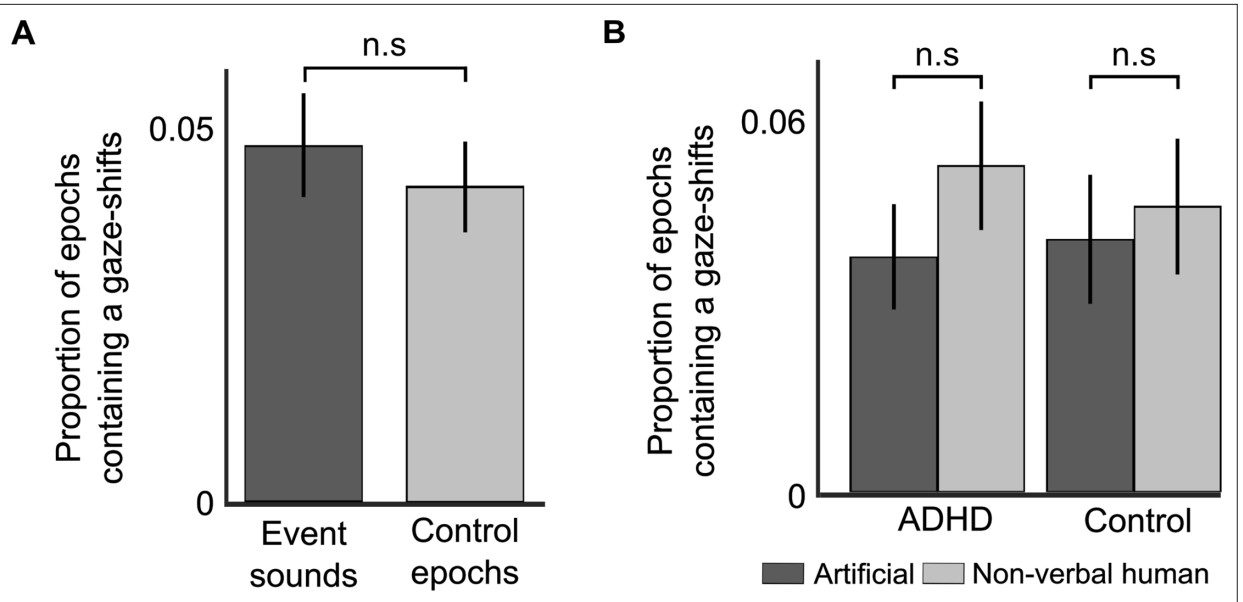

**Figure 6.** Event-related gaze-shifts. Attention deficit (hyperactivity) disorder (AD(H)D). (**A**) Bar graph showing the number of gaze-shifts performed in 2-s epochs following event-sounds versus control epochs, averaged across groups and sound types. No significant differences were found, suggesting that event-sounds were not more likely to elicit overt gaze-shifts. (**B**) Bar graph showing the number of gaze-shifts performed in 2-s epochs following each type of sound-event, separately for each group. No significant differences were found in any comparison. Error bars represent the SEM in all bar graphs.

beta-power (18–25 Hz; *Figure 7A*). Power in each frequency band was assessed for each participant at their personal peak within a pre-selected cluster of electrodes where each response was maximal (*Figure 7*). We tested whether power in either frequency differed significantly between Groups, using unpaired *t*-tests, separately for each Condition (Quiet and Event trials). However, none of the comparisons revealed any significant differences between groups, in either frequency band (*Figure 7C*) [*Alpha-power*: Events Condition: $t(47) = -0.656$, $p = 0.514$, Cohen's $d = -0.18$, $BF_{10} = 0.34$ (weak evidence for H0); Quiet Condition: $t(47) = -0.394$, $p = 0.695$, Cohen's $d = -0.11$, $BF_{10} = 0.30$ (moderate evidence for H0); *Beta-power:* Events Condition: $t(47) = -1.01$, $p = 0.315$, Cohen's $d = -0.29$, $BF_{10} = 0.43$ (weak evidence for H0); Quiet Condition: $t(47) = -0.484$, $p = 0.630$, Cohen's $d = -0.14$, $BF_{10} = 0.314$ (moderate evidence for H0)].

### Global SC levels

No significant differences between the two Groups were observed for the global tonic or phasic SC metrics, in either the Events or Quiet condition [**Phasic SC** (*Figure 7D, E*): Events Condition: $t(47) = -0.003$, $p = 0.99$, Cohen's $d = -0.0009$, $BF_{10} = 0.285$ (moderate evidence for H0); Quiet Condition: $t(47) = -0.51$, $p = 0.61$, Cohen's $d = -0.146$, $BF_{10} = 0.317$ (moderate evidence for H0); **Tonic SC**: Events Condition: $t(47) = -0.85$, $p = 0.398$, Cohen's $d = -0.244$, $BF_{10} = 0.383$ (weak evidence for H0); Quiet Condition: $t(47) = -1.65$, $p = 0.104$, Cohen's $d = -0.476$, $BF_{10} = 0.865$ (weak evidence for H0)].

### Multivariate analyses

The univariate analyses testing for between-group differences separately for each metric were complemented with a multivariate regression analysis, aimed at understanding the relative contribution of each neurophysiological metric to explaining between-group effects. For brevity, we focused on 10 key measures as described in the Methods section.

The pairwise Spearman's correlations between all measures are shown in *Figure 8*. None of the correlations were significant after correcting for multiple comparisons (fdr-correction), however three pairs did reach significance before correction (Speech-Decoding vs. ERP-P2: $r = -0.33$, $p = 0.021$; TRF-N1 vs. event-related SC: $r = -0.3$, $p = 0.033$; ERP-N1 vs. event-related SC: $r = 0.29$, $p = 0.04$). Given these generally weak correlations, we consider these measures to be independent for the purpose of multiple regression analysis.

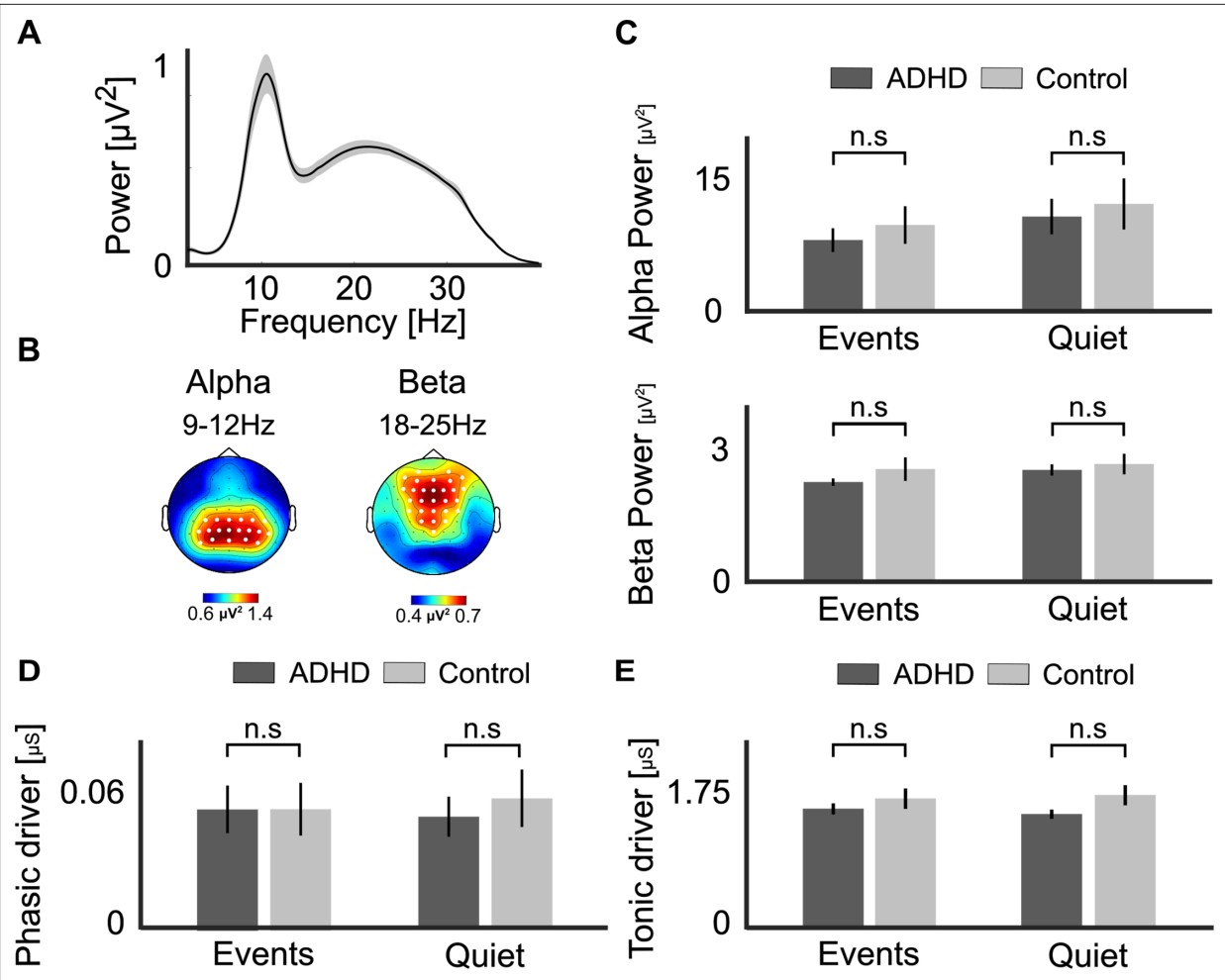

**Figure 7.** Spectral electroencephalography (EEG) and skin conductance analysis. (**A**) Grand-average power spectral density (PSD) of the periodic portion of the EEG signal (after applying the FOOOF algorithm), shows two clear peaks corresponding to the alpha (8–12 Hz) and beta (15–25 Hz) bands. Shaded areas around waveforms represent the SEM. (**B**) Topographical distribution of the average alpha- and beta-power peaks, with the clusters of electrodes used to detect personal peaks in each frequency band marked by white circles. (**C**) Average alpha- and beta-power in Group (attention deficit (hyperactivity) disorder [AD(H)D] vs. Control) and condition (Quiet and Events). (**D**) Phasic and (**E**) tonic skin conductance levels in the same groups and conditions. No significant between-group difference was found in any comparison (n.s.). Bar graphs represent the mean values, and error bars represent the SEM.

To test the contribution of each measure for predicting whether an individual was in the ADHD or control group we performed a logistic multiple regression analysis. An 'omnibus' model containing all 10 measures achieved $\chi^2$ = 15.6 with an AUC of 0.807 for predicting group allocation. We assessed the relative contribution (dominance) of each measure by comparing the AUC of the omnibus model to 10 models with a single measure held-out. *Table 1* shows ΔAUC contributed by each measure, indicating that the two most dominant measures were the ERP-N1 (positive contribution) and Speech Decoding (negative contribution), which were also the only two measures whose addition to the model reduced the AIC. This result is consistent with our univariate results, and indicates that these two measures contribute independently to between-group differences. Together, these two measures contributed an AUC of 0.15 to the omnibus model, which corresponds to half of the model's fit (Δ$\chi^2$ = 11.8, p = 0.003 relative to a model with both measures held-out).

Since ADHD is arguably not a binary category but symptoms of attentional difficulties vary on a continuum, we complemented the logistic regression analysis with a multivariate linear regression analysis, using ASRS scores as a continuous dependent measure, reflecting the severity of ADHD symptoms. An 'omnibus' model containing all 10 achieved $F(10, 38)$ = 1.48 (p = 0.18) with an $R^2$ of 0.28 for predicting ASRS scores. We assessed the relative contribution (dominance) of each measure

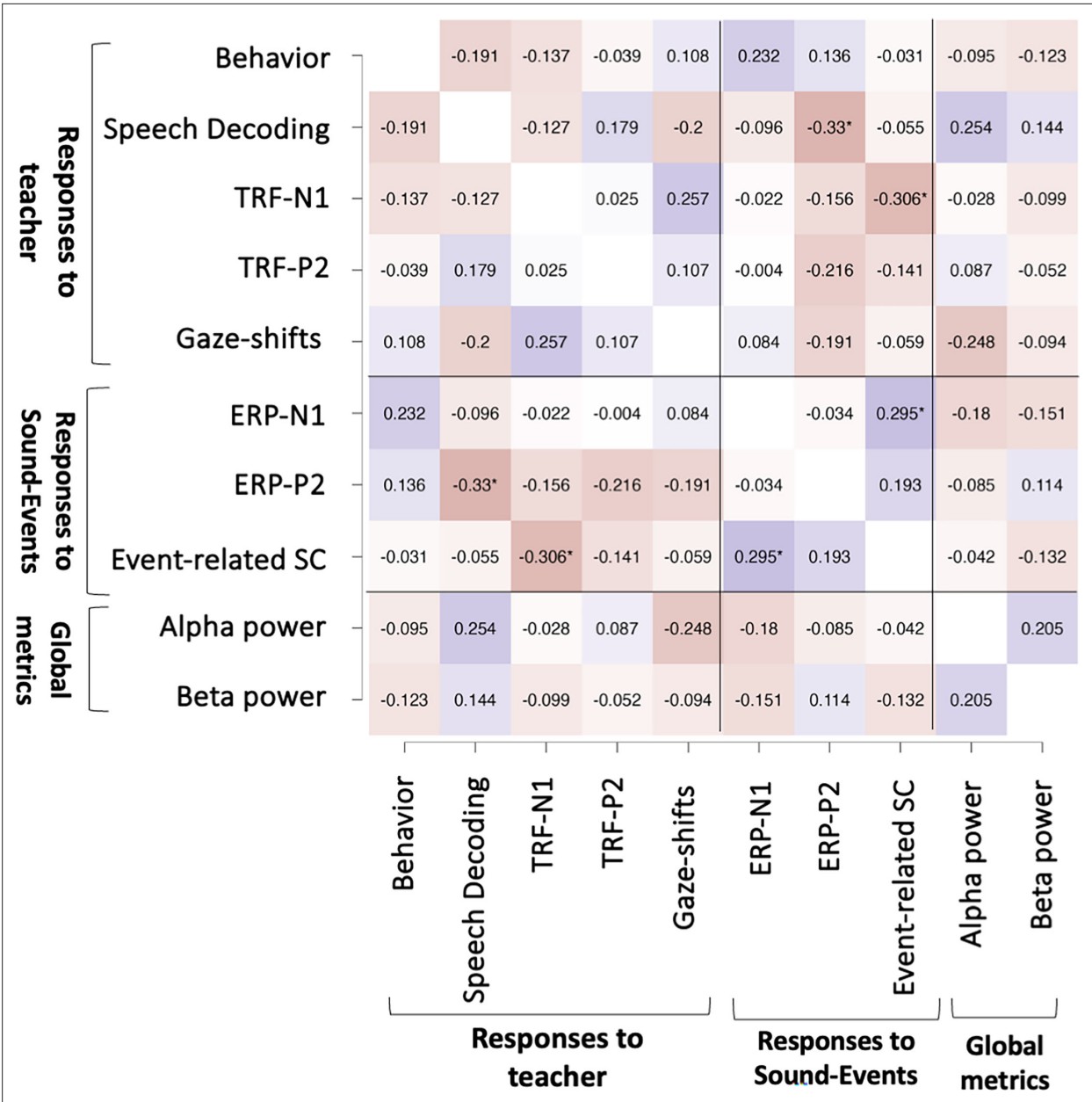

**Figure 8.** Correlation matrix. Heatmap of the pairwise Spearman's correlation coefficient between the different neurophysiological measures included in the multivariate analysis. Red-shading indicates negative correlation values and blue-shading indicate positive values. Asterisks* indicate correlation values that pass a non-corrected threshold for statistical significance, however none survived fdr-correction for multiple comparisons.

by comparing the $R^2$ in the omnibus model (containing all 10 measures) to 10 models with a single measure held-out. *Table 2* shows $\Delta R^2$ and $\Delta$AIC contributed by each measure (smaller values are better). Interestingly, this analysis identified slightly different factors than the logistic regression analysis where individuals were grouped based on having a prior diagnosis of AD(H)D (*Table 2*). The most dominant measures, who contributed at least $\Delta R^2 > 0.3$ the model, were the ERP-N1 ($\beta = 0.3$), Gaze-shifts ($\beta = 0.288$), Alpha-power ($\beta = 0.2$) and ERP-P2 ($\beta = -0.182$). Although the omnibus model was not significant (see above), a multivariate regression model that contained only the four most dominant measures did significantly predicted ASRS scores ($R^2 = 0.227$, p = 0.021), and collectively contributed $\Delta R^2 = 0.225$ (p = 0.032) to the omnibus model.

**Table 1.** Result of the dominance analysis of a multivariate logistic regression, describing the contribution of each measure for predicting whether an individual was in the ADHD or control group.

| Factor | Dominance (ΔAUC) | ΔAIC |
|---|---|---|
| ERP-N1* | 0.054 | −5.13 |
| Speech Decoding* | 0.05 | −3.29 |
| TRF-P2 | 0.02 | +0.18 |
| Beta-power | 0.01 | +1.57 |
| ERP-P2 | 0.005 | +0.48 |
| Event-related SC | −0.005 | +1.69 |
| TRF-N1 | 0.009 | +1.42 |
| Gaze-shifts | 0.007 | +1.64 |
| Alpha-power | 0.007 | +1.6 |
| Behavior | 0.007 | +1.6 |

*indicate the factors that significantly reduced the model's AIC ($p < 0.05$).

## Discussion

Recent years have seen a reckoning in the field of AD(H)D research, acknowledging that most current tools used for clinical assessments and research do not adequately capture the attentional challenges faced in real life (*Hall et al., 2016*; *Barkley, 2019*; *Mulraney et al., 2022*; *Arrondo et al., 2024*; *Schweitzer and Zion Golumbic, 2024*). We offer a new approach for advancing attention research, by utilizing VR to simulate real-life experiences together with comprehensive measurement of different neurophysiological responses. This combination affords a more objective and well-rounded description of how individuals deal with the plethora of stimuli and task demands of their environments. Here, we focused on VR classroom learning, since this is the context where many individuals, and particularly those with AD(H)D, experience difficulties in sustaining attention and avoiding distraction. Specifically, we evaluated neurophysiological measures related to focusing attention on the teacher and responses to irrelevant background sound-events alongside metrics associated more generally with levels of attention/engagement. We found that individuals diagnosed with AD(H)D exhibited significantly larger N1 response to background sound-events and somewhat reduced neural tracking of the teacher's speech. These were also the two most dominant factors contributing to a multivariate classifier that could significantly predict whether an individual was diagnosed with AD(H)D. When considering the severity of AD(H)D symptoms as a continuous variable, variance across individuals was

**Table 2.** Result of the dominance analysis of a multivariate linear regression, describing the contribution of each measure for predicting individual ASRS scores.

| Factor | Δ$R^2$ | ΔAIC |
|---|---|---|
| ERP-N1 | 0.087 | −3.58 |
| Gaze-shifts | 0.065 | −2.23 |
| Alpha-power | 0.038 | −0.49 |
| ERP-P2 | 0.033 | −0.22 |
| Behavior | 0.028 | −0.14 |
| Speech Decoding | 0.021 | +0.58 |
| TRF-N1 | 0.006 | +1.58 |
| Event-related SC | 0.001 | +1.91 |
| TRF-P2 | 0.008 | +1.18 |
| Beta-power | 0 | +1.97 |

best explained by the combined contribution of the N1 and P2 response to sound-events, amplitude of alpha-power oscillations and the frequency of gaze-shifts away from the teacher. Together, these findings emphasize the variety of attentional challenges that individuals with AD(H)D may experience in realistic scenarios and provide novel insights into the underlying neurophysiological mechanisms.

## Neurophysiological responses in the VR classroom associated with symptoms of AD(H)D

The finding with the largest effect size in this study was larger N1 response to background sound-events in individuals with AD(H)D. This response showed a significant univariate between-group effect, and was the most dominant contributor to predicting group-classification (AD(H)D vs. control) as well as the severity of AD(H)D symptoms. Enhanced N1 responses are often associated with heightened sensory sensitivity (*Heil, 1997*; *Lütkenhöner and Klein, 2007*), perceptual predictability (*Chennu et al., 2013*; *Lange, 2013*; *Schwartze and Kotz, 2015*), and may reflect a transient detector mechanism of surprising events that can potentially (though not necessarily) trigger an attention shift (*Näätänen and Picton, 1987*; *Escera et al., 1998*; *Escera et al., 2003*; *Berti, 2013*). Interestingly, most studies comparing auditory ERPs in individuals with and without AD(H)D, using more traditional paradigms (e.g., oddball), typically *have not reported* differences in early-sensory components such as the N1, but rather sometimes find group differences for later ERP responses (*Barry et al., 2003*; *Gumenyuk et al., 2005*; *Kaiser et al., 2020*; *Peisch et al., 2021*). As discussed below, we attribute these discrepancies to the non-ecological nature of most traditional paradigm. The current result is in line with increased sensory sensitivity to background events, which might be especially pertinent under ecological condition. In addition, we found that neural tracking of the teacher's speech was reduced in the AD(H)D group, and this metric also emerged as a dominant predictor of AD(H)D diagnosis in multivariate between-group classification. Neural speech tracking of continuous speech is known to be modulated by attention, and is reduced when speech is not attended (*Ding and Simon, 2012b*; *Mesgarani and Chang, 2012*; *Zion Golumbic et al., 2013*; *Petersen et al., 2017*; *Vanthornhout et al., 2019*; *Kaufman and Zion-Golumbic, 2023*). We are not aware of previous studies that directly investigated neural speech tracking in AD(H)D, however these results are consistent with reduced levels of sustained attention toward the teacher in this group.

Reduced sustained attention and higher proneness to distraction are, of course, considered key characteristics of AD(H)D, however are rarely demonstrated under realistic conditions, using ecological tasks and materials. The current results demonstrate how, by coupling the ecological validity and flexibility of VR with recordings of neural activity, we can substantially advance our mechanistic understanding of attention and distractibility and obtain objective metrics of stimulus processing in realistic scenarios. Interestingly, since the magnitude of the N1 to sound-events and speech tracking of the teacher were not correlated with each other, these results do not support the notion of an inherent 'tradeoff' between paying attention to the teacher and responding to irrelevant stimuli. Rather, they may represent two independent ways in which individuals with AD(H)D differ from their peers in the manner they respond to and process stimuli in dynamic environments.

However, adopting a binary perspective of AD(H)D can be misleading, as it does not capture the vast variability within groups, as emphasized by the RDoC framework (*Morris et al., 2022*). This is evident from inspecting the distribution of ASRS scores in the current sample, which shows that although the two groups are clearly separable from each other, they are far from uniform in the severity of symptoms experienced. For this reason, we conducted a continuous multiple regression analysis, to identify which neurophysiological measured explained variance in self-reported AD(H)D symptoms. This analysis confirmed the link between the magnitude of the N1 response to sound-events to AD(H)D symptoms, but also identified several additional factors that contribute to explaining variance in ASRS scores across individuals, primarily: the level of alpha-power and the frequency of gaze-shifts away from the teacher. This finding is consistent with the hypothesized role of these metrics in attention. Higher alpha-power is associated with reduced levels of attention/arousal, and is consistently found to increase in conditions of boredom, tiredness or prolonged time on tasks (*Clarke et al., 2001*; *Dockree et al., 2007*; *Palva and Palva, 2007*; *Wöstmann et al., 2017*; *Boudewyn and Carter, 2018*; *Jin et al., 2019*; *Haro et al., 2022*). Several studies have also found higher baseline alpha-power in individuals with AD(H)D versus controls (*Barry et al., 2003*; *Johnstone et al., 2013*; *Bozhilova et al., 2022*; *Michelini et al., 2022*), although results are not always consistent (*Loo and Makeig, 2012*;

*Johnstone et al., 2013*). Similarly, gaze-shifts are often used as a proxy for distraction, particularly under naturalistic conditions (*Marius 't Hart et al., 2009*; *Foulsham et al., 2011*; *Risko and Kingstone, 2011*; *Risko et al., 2012*; *Hoppe et al., 2018*), and recent VR and real-life eye-tracking studies have found increased frequency and prolonged duration of gaze-shifts toward irrelevant locations in individuals with AD(H)D (*Braga et al., 2016*; *Türkan et al., 2016*; *Vakil et al., 2019*; *Mauriello et al., 2022*; *Stokes et al., 2022*). The current results extend these findings to more ecological contexts and indicate that individuals who exhibited higher alpha-power, performed more frequent gaze-shifts and had larger N1 response to background sound-events in our VR classroom were more likely to report experiencing attentional difficulties in real life.

It is important to note that although the current results had significant explanatory power of variance in AD(H)D symptoms, they are far from exhaustive with substantial variance still unexplained by these neurophysiological measures. In this regard it is important to bear in mind that the measures of AD(H)D themselves (diagnosis and ASRS scores) are also not entirely reliable, given the diversity diagnostic practice (*Hall et al., 2016*; *Barkley, 2019*; *Mulraney et al., 2022*; *Arrondo et al., 2024*; *Schweitzer and Zion Golumbic, 2024*) and the subjective nature of self-reports which are prone to bias (e.g., recency, salience, and generalization effects; *Rabiner et al., 2010*; *Toplak et al., 2013*; *Brevik et al., 2020*). What is clear, though, is that no single neurophysiological measure alone is sufficient for explaining differences between the individuals – whether through the lens of clinical diagnosis or through report of symptoms. These data emphasize the complex nature of attentional functioning, especially under realistic conditions, and that fact that 'attentional challenges' may manifest in a variety of different ways, indexed though different neurophysiological responses. Moving forward, we believe that by measuring multiple aspects of neural and physiological responses, and studying these in diverse and ecologically valid conditions, will ultimately lead to distilling specific neurophysiological patterns that can reliably generalize across contexts.

## What about the P300?

Notably, the current results diverge from previous studies of distractibility in AD(H)D in several ways. In traditional ERP studies, distractibility by irrelevant stimuli is often associated with a family of late-latency positive neural responses (the P300 family), and specifically the P3a ERP component, which is typically observed at a latency of ~250–400 ms after hearing novel or surprising sounds (*SanMiguel et al., 2010*; *Wetzel et al., 2013*; *Masson and Bidet-Caulet, 2019*; *Barry et al., 2020*). Given this vast literature, we had assumed that background sound-events in our VR classroom would also elicit this response. Moreover, past studies have shown reduced P3a responses in individuals with AD(H)D (*Barry et al., 2003*; *Gumenyuk et al., 2005*; *Kaiser et al., 2020*; *Peisch et al., 2021*; *Kwasa et al., 2023*), leading us to expect a similar pattern here. However, here the early N1 response to sound-events was larger in the ADHD group but no differences were found for any of later responses. Moreover, visual inspection of the ERPs to Artificial and Non-verbal human sound-events illustrates that not all ecological sounds elicit the same time-course of responses in mid- to late time-windows.

When comparing the current results from the VR classroom with more traditional ERP studies of distractibility, we must consider the vast differences in context and task requirements and the 'functional status' of background sound-events in these studies. The P3a is typically observed in tasks that require high levels of perceptual vigilance to discriminate between specific stimulus features, such as the pitch, duration or intensity of sounds presented in an oddball paradigm. In these tasks, so-called 'novel' or 'distractor' sounds/stimuli are presented as part of the sequence of stimuli that need to be discriminated, and these stimuli elicit a P3a (whereas target sounds elicit a somewhat similar P3b response; *Barry et al., 2020*). Accordingly, although these 'distractor' stimuli do not require an explicit response, they are in fact within the focus of participants' endogenous attention, rather than truly 'background' stimuli (*Makov et al., 2023*). This differs substantially from the role of the novel sounds-events in the VR classroom task used here, where sound-events are clearly in the background– both spatially and semantically – and occur in parallel to an ongoing speech stimulus (the lecture) that participants are primarily paying attention to. In addition, the perceptual and cognitive demands of the VR Classroom – where participants are instructed to listen and understand the speech as they would in a real-life classroom – differ substantially from the highly vigilant demands of speeded-response tasks. These important differences in task and stimulus characteristic ultimately lead to a different treatment of novel sound-events by the brain, emphasizing the importance of considering

context when comparing results across experimental designs (*Makov et al., 2023*; *Mandal et al., 2024*).

## Null results – insufficient sensitivity or true lack of differences?

Besides the lack of a reliable P300-like effect here, several additional measures that were evaluated here did not show any group-wise effects and did not contribute to explaining ASRS scores, despite previous literature suggesting that they might. Although null results are difficult to interpret fully, we find it worthwhile to note these here and briefly discuss potential reasons for the lack of group-related effects. Perhaps the most surprising null effect was the lack of differences in behavioral performance. Participants in both groups achieved similar behavioral performance in the Quiet and Events condition, suggesting that the presence of occasional sound-events ultimately did not ultimately impair their ability to understand the content of the mini-lectures (at least at the level probed by our comprehension questions). These results should be interpreted bearing in mind the details of the current task design – we deliberately chose a task that would mimic the level of information processing required in real-life learning contexts as opposed to speeded-response tasks used in traditional attention research that encourage failures of performance by creating perceptually 'extreme' conditions (*Hall et al., 2016*; *Barkley, 2019*; *Arrondo et al., 2024*). Moreover, the presence of occasional sound-events – while noticeable, was not design to be extremely adverse. Under these conditions, individuals with and without AD(H)D were able to maintain good performance despite the occasional acoustic disturbances (*Mandal et al., 2024*), indicating that neurophysiological indication of somewhat reduced attention do not necessarily translate to poorer behavioral outcomes. It is of course possible that larger behavioral effects would have emerged if task demands or the severity of acoustic disturbances were more challenging.

Besides neural evoked responses, background sound-events also elicited a transient increase in SC that is associated with an 'orienting reflex', reflecting transient changes in arousal/alertness (*Frith and Allen, 1983*; *Zimmer and Richter, 2023*). This response is considered obligatory and automatic, and does not require allocation of attention as it can be observed during sleep and other unconscious states (albeit the magnitude of these responses can vary as a function of attention). Although we had hypothesized that this response might be heightened in individuals with AD(H)D, as has been reported in some studies (*Sergeant, 2000*; *Bellato et al., 2020*) (similar to the effect on the N1 response), the current results do not support this. Indeed, there has been much debate regarding the relationship between the SC orienting reflex and early neural responses, with conflicting results reported across studies (*Lim et al., 1996*; *MacDonald and Barry, 2020*). Additional research is likely required to better characterize the relationship between neural and physiological responses to surprising events in the environment.

Last, another interesting null result was the lack of overt-gaze-shifts following sound-events. Gaze-shifts are often considered a proxy for distractibility and capture of attention (*Parkhurst et al., 2002*; *Geisler and Cormack, 2011*; *Nissens et al., 2017*), particularly in individuals with AD(H)D. Accordingly, we had expected to find that sound-event elicited overt shifts toward the spatial location from which they were emitted. However, this was not the case, and individuals (in both groups) did not perform more gaze-shifts following sound-events relative to their baseline tendencies. A similar result was recently reported by our group in a VR café scenario, where that hearing semantically salient words in background speech (e.g., your name or semantic violations) did elicited neural and SC responses, but was not accompanied by gaze-shifts (*Brown et al., 2023*). These results might indicate that peripheral auditory stimuli are, generally, less effective than visual stimuli at capturing overt attention (*Turoman and Vergauwe, 2024*; *Mandal et al., 2024*), or that in ecological audiovisual settings individuals are less prone to move their eyes to search for the source of auditory disturbances, despite noticing them. These possibilities will be further investigated in follow-up VR studies, comparing the effects of irrelevant auditory and visual events.

## General considerations

Before concluding, we note several important caveats, pertaining to the specific sample tested here. First, as noted, classification of the AD(H)D group was based on their prior clinical diagnosis, which likely varied across participants in terms of diagnosis approach and age of diagnosis. Although the two groups were distinguishable in their self-reported ASRS scores, a more reliable assessment of

the severity of current AD(H)D symptoms might have been achieved if we had conducted a full clinical assessment of all participants, as per the acceptable clinical guidelines (*American Psychiatric Association, 2022*). Second, this study was conducted on adults, whereas the majority of AD(H)D research – and particularly all VR classroom studies – have focused on children (*Pollak et al., 2009*; *Neguţ et al., 2017*; *Schweitzer and Rizzo, 2022*; *Seesjärvi et al., 2022*; *Stokes et al., 2022*). AD(H)D is primarily defined as a childhood condition and although it often persists into adulthood, with profound implications on professional and academics outcomes, how the nature of the deficit changes with age is still unknown (*Barkley et al., 2008*). Moreover, the adult cohort tested here included primarily university students, who do not represent all adults but rather are a subgroup with relatively high cognitive and learning abilities, who are accustomed to sitting in lectures for long periods of time (*Henrich et al., 2010*). Hence, it is possible that the lack of group differences in some metrics in the current dataset (and particularly in their behavioral outcomes) is because these individuals have learned to deal with distraction and have developed appropriate strategies, which allow them to thrive in academic settings. This caveat highlights the importance of considering sampling techniques in group-comparison studies, which sometimes limits their interpretation to more circumscribed subsamples (*Rad et al., 2018*). At the same time, even if the current findings pertain only to this highly functioning academic subgroup of individuals with AD(H)D, they imply an impressive ability to adapt and cope with the naturally occurring distractions of realistic classrooms. Specifically, they suggest that withstanding distraction in real-life contexts may be improved through training and good habit-forming, even in individuals with AD(H)D. We hope to follow up on this research in the future, to deepen our understanding of the diversity within the AD(H)D and typical population and to better characterize attention and distractibility across different types of real-life scenarios.

## Conclusions

Paying attention in class requires both investing cognitive resources in processing the lesson itself (e.g., the teacher's speech), and trying to minimize intrusions from background events. Here, we found that both of these operations are less effective in individuals who are diagnosed with AD(H)D or who report more severe symptoms of difficulties in everyday attention. Specifically, these individuals exhibit heightened sensory response to irrelevant sounds, reduced speech tracking of the teacher, a tendency toward more frequent gaze-shifts away from the teacher, and somewhat increased alpha-power. Interestingly, since these metrics were not correlated with each other, they likely represent different ways through which inattention or distraction may manifest, rather than a general 'biomarker' of inattention. Counter to some prevalent notions, these data suggest that AD(H)D may not be characterized by a specific 'neural marker' indicating attentional deficits, particularly when tested under ecological circumstances (*Barkley, 2019*; *Faraone et al., 2021*; *Sonuga-Barke et al., 2023*). Rather, they resonate with recent calls to revisit the parameters used for defining 'attention-disorders' and the need to adopt more nuanced and dimensional approach, as well as designing more ecologically valid means of testing, in order to better characterize and understand the complex nature of real-life attentional challenges (*Marcus and Barry, 2011*; *Heidbreder, 2015*; *Morris et al., 2022*; *Elahi et al., 2024*).

## Materials and methods

### Participants

Fifty-four participants initially participated in the study; however, the data from five participants were excluded due to technical issues or their voluntary request to discontinue participation. The final sample consisted of 49 participants (21 male, 28 female; 46 right-handed, 3 left-handed). Of these participants, 24 had a prior clinical diagnosis of attention deficit hyperactivity disorder (AD(H)D group), and 25 participants did not (control group). The two groups were matched for age, which ranged between 20 and 38 years (mean age of all participants: 26.9 ± 3.0; AD(H)D participants: 26.7 ± 3.7; control participants: 25.7 ± 4.4). The majority of participants were university students. Participants in the AD(H)D group were questioned about their medication intake regimen, and the experiment was scheduled at least 12 hr after their last dose of medication. None of the participants (in either group) reported taking any other medications on a regular basis. The experimental protocol was approved by the ethics committee at Bar-Ilan University (protocol # ISU202112002), and all participants provided

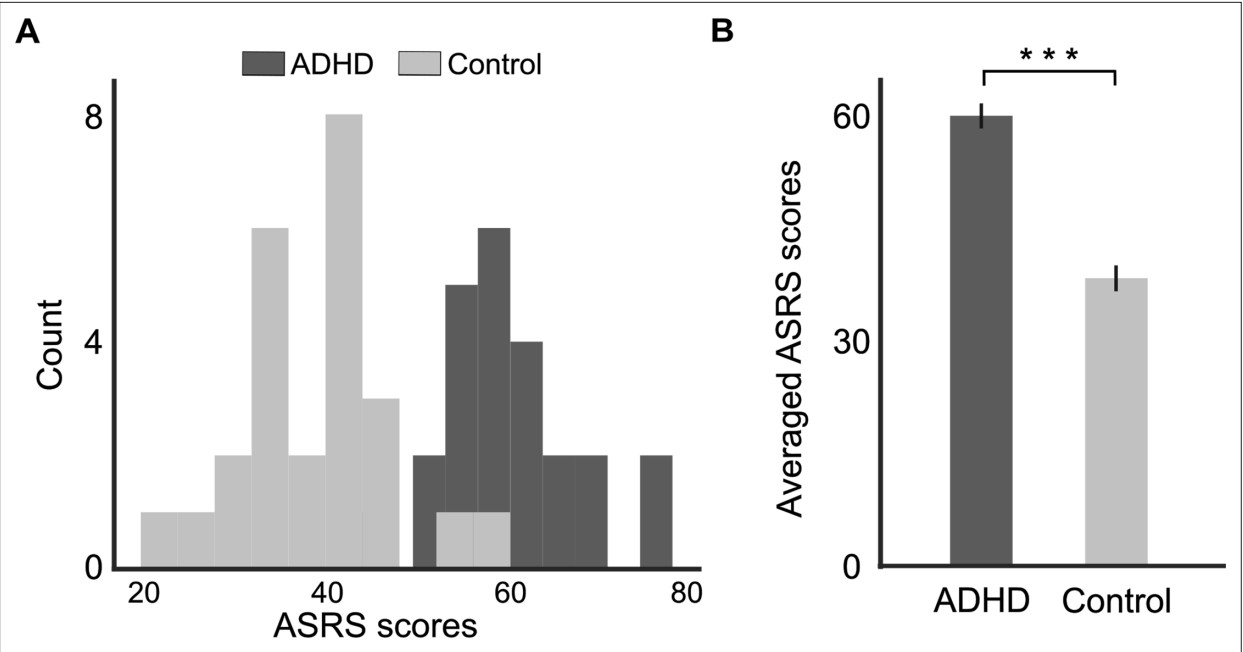

**Figure 9.** Histogram (**A**) and bar graph (**B**) showing the distribution of ASRS scores in the attention deficit (hyperactivity) disorder (AD(H)D) and Control groups. Error bars represent standard error of the mean (SEM). This confirms group allocation, within the tested sample.***p < 0.001.

written informed consent prior to their involvement in the study and data collection procedures. Participants received financial compensation or course credit in exchange for their participation.

## ASRS questionnaire

All participants completed a self-report questionnaire assessing AD(H)D symptoms (ASRS-v1.1; *Adler et al., 2006*, Hebrew version: *Zohar and Konfortes, 2010*), which has been shown to have good psychometric properties high sensitivity for AD(H)D diagnosis (*Kessler et al., 2005*). Participants in the AD(H)D group were instructed to complete the questionnaire based on their experiences at a time when they are <u>not</u> taking medication. The questionnaire consists of 18 items, which are rated on a 5-point Likert scale ranging from 0 (never) to 4 (very often). ASRS scores for each participant were calculated as the total sum of marks across all items, as this has been shown to yield the highest sensitivity to AD(H)D diagnosis (*Zohar and Konfortes, 2010*). We did not try to categorize AD(H)D subtypes (e.g., inattention, hyperactivity, and combined) since these subtypes are less reliable in adulthood. As expected, the group diagnosed with AD(H)D exhibited significantly higher ASRS scores compared to the control group [$t(47) = 8.49$, $p < 0.001$; see *Figure 9*]. This outcome validates the group allocation and demonstrates the consistency between AD(H)D diagnosis and self-reported AD(H)D symptoms.

## The VR classroom: design and stimuli

The VR classroom platform was developed using the Unity game engine (unity.com; JavaScript and C# programming). Development and programming of the VR classroom were done primarily in-house, using assets (avatars and environment) were sourced from pre-existing databases. The classroom environment was adapted from assets provided by Tirgames on TurboSquid (https://www.turbosquid.com/Search/Artists/Tirgames) and modified to meet the experimental needs. The avatars and their basic animations were sourced from the Mixamo library, which at the time of development supported legacy avatars with facial blendshapes (this functionality is no longer available in current versions of Mixamo). A brief video example of the VR classroom is available at https://osf.io/svjqg.

Participants experienced the virtual environment through an HTC Vive ProEye VR headset that is equipped with an embedded eye-tracker (120 Hz binocular sampling rate). Audio was delivered through in-ear headphones (Etymotic ER1-14 with disposable foam eartips).

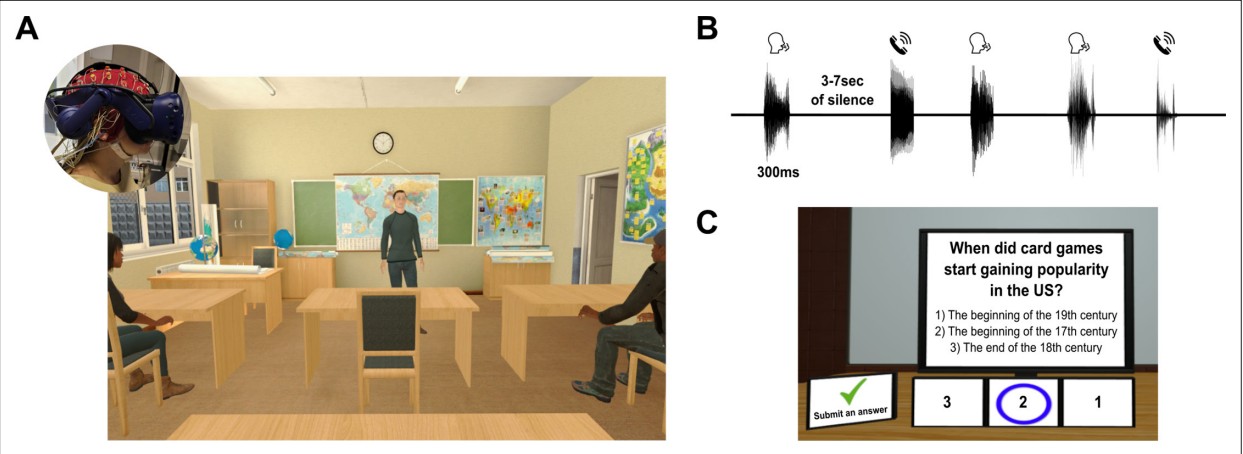

**Figure 10.** Virtual reality classroom setup. (**A**) Illustration of the virtual classroom used in the study.Inset: a participant wearing the virtual reality (VR) headset over the electroencephalography (EEG) cap. (**B**) Example of the sequence of sound-events presented in Event-trials, which were emitted from a spatial location to the right or left of the participant. Event-trials contained five events, of both types (Artificial and Human non-verbal sounds), randomized across trials and separated by 3–7 s. (**C**) Example of a multiple-choice comprehension question presented after each trial (English translation).

In the VR classroom (*Figure 10A*), participants experienced sitting at a desk in the second row of a classroom, facing a male avatar-teacher, who was standing in front of a blackboard and a wall adorned with maps. Ten additional avatar-students occupied the remaining desks in the classroom. Although these student avatars did not engage in verbal communication during the experiment, their bodies were animated to simulate natural sitting movements. The avatar-teacher delivered a series of 30 mini-lessons on a range of scientific and historical topics (mean duration 42.7 s ± 5.1). The audio for these mini-lessons was taken from existing informational podcast, recorded by a male speaker. Speech audio was presented using 3D spatial audio (implemented in Unity) to be perceive as emitting from the teacher's lips. To achieve realistic lip-speech synchronization, the teacher's lip movements were controlled by the temporal envelope of the speech, adjusting both timing and mouth size dynamically. His body motions were animated using natural talking gestures.

Besides the teacher's lesson, short sounds (sound-events) were presented occasionally in most trials, from a spatial location to the right or left of the participant (randomized), as if originating from one of the students sitting beside them in class (*Figure 10B*). A total of 110 different sounds were used, taken from existing sound databases (freesfx.co.uk, findsounds.com, freesound.org), and the IADS-E sound stimuli database (*Yang et al., 2018*), from two broad categories: Non-verbal human sounds (mainly coughs and throat clearings), and Artificial sounds (phone ringtones and digitally generated 'boing' sounds). All sound-events were normalized to the same loudness level using the Normalize function in the audio-editing software Audacity (theaudacityteam.org, ver 3.4), with the peak amplitude parameter set to –5 dB, and trimmed to a duration of 300 ms.

In the VR classroom, the loudness of the teacher's speech was set to a comfortable level for each participant, as determined in a training trial. The precise intensity level cannot be measured reliably, since it is affected by the specific positioning of the foam earphone inserts in the participants' ear, but was roughly between 60 and 70 dB SPL. The sound-events were played at a loudness level of 50% relative to the teacher's voice (–3 dB SPL).

## Experimental procedure

The experiment was conducted in a sound attenuated room. After comfortably fitting the VR headset on the participant, the eye-tracker was calibrated and validated using a standard 9-point calibration procedure. Participants were then familiarized with the VR classroom scene, and performed two training trials – one without events-sounds and one with – for training and calibration purposes.

The experiment consisted of 30 trials. In each trial, the avatar-teacher delivered a different mini-lesson and participants were instructed to pay attention to its content. They were not given any specific instructions regarding the other sounds or stimuli in the classroom. After each mini-lesson,

participants answered four multiple-choice questions about its content (*Figure 10C*). Most trials (22) contained occasional background event-sound (Events condition; with 5 of both types sound-events per trial, separated by 3–7 s; *Figure 10B*), whereas 8 trials did not, serving as a Quiet baseline condition. In total, 104 sound-events were presented throughout the experiment (52 of each type). The order of mini-lecture presentation and their allocation to the Events or Quiet condition was randomized across participants.

## EEG, GSR, and eye-gaze recordings

Neural activity was recorded using 64 active ag-agCl EEG electrodes (BioSemi) and was sampled at 1024 Hz. Electrooculographic (EOG) signals were also included in this recording, measured by three additional electrodes located above the right eye and on the external side of both eyes.

The SC response (*Akash et al., 2018*) was recorded using two passive Ni hon Kohden electrodes placed on the fingertips of the index and middle fingers of participants' non-dominant hand. These signals were also sampled by the BioSemi system, and thus synchronized to EEG data.

Eye-gaze data were recorded using the eye tracker embedded in the VR headset, which provides information about the *x–y–z* coordinate of the gaze in 3D space at each time-point, as well as automatic blink-detection. We defined several a priori regions of interest (ROIs) in the 3D classroom space, which included: 'Teacher', 'Left Board', 'Right Board', 'Ceiling', 'Floor', 'Middle Window', 'Left Student', 'Right Student'. Note that the sound-events were emitted from the ROI labeled 'Right Student'. Using a combined calculation of the participants' head position and direction of the gaze vector, gaze data could be automatically classified to describe which ROI the participant looked at, at each point in time.

## Behavioral data analysis

To evaluate participants' comprehension and memory retention, we calculated their accuracy on four multiple-choice questions following each trial (average correct responses). Between-group differences were assessed using unpaired *t*-tests, separately for the Quiet and Events conditions. These were evaluated separately (here and in all other analyses), given the imbalanced number of trials in each condition. The $BF_{10}$ was also calculated for each pairwise comparison, to test the level of confidence for embracing or rejecting the null hypothesis (H0; $BF_{10}$ thresholds used throughout: $BF_{10} < 1$ indicates weak support of H0 and $BF_{10} < 0.25$ indicates moderate/strong support of H0; $BF_{10} > 1$ indicates weak support for rejecting H0 and $BF_{10} > 2.5$ indicates moderate/strong support for rejecting H0).

## EEG data analysis

The EEG data were preprocessed using MATLAB with the FieldTrip toolbox (https://www.fieldtrip-toolbox.org, version 20220729). The raw EEG data was referenced to linked right and left mastoids, then band-pass filtered between 0.5 and 40 Hz (fourth-order zero-phase Butterworth IIR filter), detrended and demeaned. Visual inspection was performed to identify and remove gross artifacts (excluding eye movements). Independent component analysis was then used to further remove components associated with horizontal or vertical eye movements and heartbeats, which were selected manually based on their characteristic time-course patterns and topographical distributions, consistent with known artifact signatures. Remaining noisy electrodes, containing extensive high-frequency activity or DC drifts, were interpolated using adjacent electrodes, either on the entire dataset or on a per-trial basis, as needed.

Three types of analyses were applied to the clean EEG data: (1) Speech tracking analysis of the neural response to teacher's speech, (2) ERPs analysis capturing the time-locked neural response to the Sound-events, and (3) Spectral analysis, looking specifically at ongoing power in two canonical frequency bands previously associated with attention: the alpha (6–15 Hz) and beta (13–30 Hz) ranges.

### Speech tracking response (TRF)

The clean EEG data were segmented into trials from the onset the teacher's speech until the end of the trial. For this analysis, EEG data were further bandpass filtered between 0.8 and 20 Hz (fourth-order zero-phase Butterworth IIR filter), and downsampled to 100 Hz for computational efficiency. To ensure comparability across participants and conditions, the EEG data were normalized using *z*-score transformation. To estimate the neural response to the speech we performed speech-tracking

analysis which estimates a linear TRF that best describes the relationship between features of the stimulus (S) and the neural response (R) in a given trial. Speech tracking analysis was conducted using the mTRF MATLAB toolbox (*Crosse et al., 2016*), using both an encoding and decoding approach. Speech-tracking analysis was limited only to the Events condition, since the Quiet condition contained substantially less data (eight trials; ~5 min of data in total), which was insufficient for obtaining a reliable speech-tracking response (predictive power non-significant vs. permutations of shuffled data, p = 0.43). For speech tracking in the Events condition we used a multivariate approach, and included the temporal envelopes of both the teacher's speech and the sound-events as separate regressors (S) when modeling the neural response, so as to represent the entire soundscape they heard. For both regressors, we derived a broadband temporal envelope by filtering the audio through a narrow-band cochlear filterbank, extracting the narrowband envelopes using a Hilbert transform, and averaging across bands. The regressors were then downsampled to 100 Hz to match the EEG data and normalized using *z*-score.

In the **encoding** approach, TRFs are estimated for each EEG channel, with lags between the S and the R ranging from –150 ms (baseline) to 450 ms. The mTRF toolbox uses a ridge-regression approach for L2 regularization of the model to ensure better generalization to new data. We tested a range of ridge parameter values ($\lambda$'s; between $10^{-2}$ and $10^4$) and used a leave-one-out cross-validation procedure to assess the model's predictive power, whereby in each iteration, all but one trials are used to train the model, and it is then applied to the left-out trial. The predictive power of the model (for each $\lambda$) is estimated as the Pearson's correlation between the predicted neural responses and the actual neural responses, separately for each electrode, averages across all iterations. We report results of the model with the $\lambda$ the yielded the highest predictive power at the group-level (rather than selecting a different $\lambda$ for each participant which can lead to incomparable TRF models across participants; see discussion in *Kaufman and Zion-Golumbic, 2023*).

The statistical reliability of the model's predictive power at each electrode is evaluated using a permutation test where the same procedure is repeated 100 times on shuffled data where the S from each trial is randomly paired with the R from a different trial. From each permutation, we extract the highest predictive power across all electrodes, yielding a null distribution of the maximal predictive power that could be obtained by chance. The predictive power in the real data is compared to this distribution, and electrodes with values falling in the top 5%$^{tile}$ of the null distribution are considered to have a significant speech tracking response. Note that by choosing the maximal value across electrodes in each permutation, this procedure also corrects for multiple comparisons.

A complementary way to examine the neural representation of continuous speech is the **decoding** approach. In the decoding model, a multidimensional transfer function is estimated using the neural data (R) from all electrodes as input in attempt to reconstruct the stimulus feature (S).

For the decoding analysis we used time lags between S and R ranging from –400 to 0 ms (negative lags imply that the S precedes the R; here including a baseline would be detrimental to decoding). Here too, we used a ridge-regression regularization approach and used a leave-one-out cross-validation procedure to assess the quality of the reconstruction. We tested a range of ridge parameter values ($\lambda$'s; between $10^{-2}$ and $10^4$) and used a leave-one-out cross-validation procedure to assess the model's predictive power, whereby in each iteration, all but one trials are used to train the model, and it is then applied to the left-out trial. The quality of the model (for each $\lambda$) is estimated as the Pearson's correlation between the reconstructed S of a left-out-trial vs, the actual S (reconstruction correlation). We report results of the model with the $\lambda$ the yielded the highest predictive power at the group-level. The statistical significance of the reconstruction correlation was assessed using a permutation test in which the same procedure is repeated 100 times on shuffled data where the S from each trial is randomly paired with the R from a different trial. This yields a null distribution of predictive power values that could be obtained by chance, and the real model is considered reliable if its predictive power falls within the top 5%$^{tile}$ of the null distribution.

Speech tracking analysis – using both encoding and decoding approaches – was performed separately for each Group (AD(H)D and Control). To test for between-group differences, we compared the models' predictive power, using unpaired *t*-tests. We additionally calculated the $BF_{10}$ for each comparison, to test the level of confidence for embracing or rejecting the null hypothesis.

## Event-related response

To quantify neural responses to the sound-events, we estimated the time-locked ERPs. For this analysis, the clean EEG data were segmented into epochs ranging from −100 to 500 ms around each event. The data were further low-passed filtered at 12 Hz, as is common in ERP analysis (fourth-order zero-phase Butterworth IIR filter), and baseline corrected to the pre-stimulus period (−100 to 0 ms). ERPs were derived separately for each group (AD(H)D vs. Control) and Event-type (Artificial vs. Non-verbal human sounds). Visual inspection of the ERPs showed that they were dominated by two early components: the N1-component (75–200 ms) and later P2-component (210–260 ms), which primarily reflect auditory sensory responses, but are sometimes modulated by attention (*Hillyard et al., 1973*; *Woldorff and Hillyard, 1991*).

Statistical analysis of the ERPs focused on the two simple effects, using a Monte-Carlo spatio-temporal cluster-based permutation test, across all electrodes and time-points. To test for difference in the ERPs between Groups, we used unpaired *t*-tests on the ERPs, averaged across Event-types. For this analysis, we had to set the cluster alpha-level to a very low level of $p = 10^{-5}$, in order to obtain separate clusters for the N1 and P2 responses. To test for difference in the ERPs generated for the two Event-types, we use paired *t*-tests on the ERPs of all participant, irrespective of their group. For this analysis, the cluster alpha-level was set to $p = 0.05$. Last, to verify the main effects and in order to test for potential interactions between Group and Event-type, we extracted the peak amplitudes of the N1 and P2 component from each participant, and performed a mixed ANOVA with repeated measures, with the factors Group (AD(H)D vs. Control; between) and Event-Type (Artificial vs. Human non-verbal; within).

## Spectral analysis

The third analysis performed on the EEG data was a spectral analysis. This analysis was performed on the clean EEG data, segmented into full trials (same segmentation used for the speech-tracking analysis). We calculated the EEG spectral power density (PSD) of individual trials using multitaper fast-fourier transform (FFT) with Hanning tapers (method 'mtmfft' in the fieldtrip toolbox). The PSDs were averaged across trials for each participant, separately for each electrode and condition (Quiet and Events). We used the FOOOF algorithm (*Donoghue et al., 2020*) to decompose the PSD into periodic (oscillatory) and aperiodic components. The periodic portion of the PSD showed clear peaks in the alpha (8–12 Hz) and beta (15–25 Hz) ranges, suggesting reliable oscillatory activity (*Figure 7*). Since the peaks of alpha and beta activity can vary substantially across individuals, for each participant, we identified the frequency with the largest amplitude within each range, focusing on a cluster of occipital–parietal electrodes (for alpha) or frontal–central electrodes (for beta), where these responses were largest. These were used for statistical analyses of between-group differences. assessed using unpaired *t*-tests, separately for the Quiet and Events conditions, in each frequency band, and the $BF_{10}$ was calculated for each pairwise comparison to test the level of confidence for embracing or rejecting the null hypothesis.

## SC data analysis

The SC signal was analyzed using the Ledalab MATLAB toolbox (version 3.4.9; *Benedek and Kaernbach, 2010*; http://www.ledalab.de/). The raw data were downsampled to 16 Hz using FieldTrip's ft_resampledata function, which applies a built-in anti-aliasing low-pass filter to prevent aliasing artifacts. Data were inspected manually for any noticeable artifacts (large 'jumps'), and if present were corrected using linear interpolation in Ledalab. A continuous decomposition analysis (CDA) was employed to separate the tonic and phasic SC responses for each participant. The CDA was conducted using the 'sdeco' mode (signal decomposition), which iteratively optimizes the separation of tonic and phasic components using the default regularization settings. We tested for between-group differences in *global SC levels* (tonic and phasic responses separately, averaged across full trials) using unpaired *t*-tests, separately for the Quiet and Events conditions and the $BF_{10}$ was calculated for each pairwise comparison.

In addition, we analyzed *event-related changes in SC* following Sound-events. This analysis focused on a time-window between 0 and 5 s after the onset of Sound-events. Since changes in SC are relatively slow and peak between 2 and 3 s after a stimulus, the mean response between 0 and 1 s was used as a baseline and was subtracted from the signal. Statistical analysis focused the time-window

surrounding the mean response between 2 and 3 s, extracted for each participant. A mixed ANOVA with repeated measures was used to test for main effects of Group (AD(H)D vs. Control; between factor), and Event-Type (Artificial vs. Non-verbal human; within factor) and the interaction between them.

## Eye tracking data analysis

Eye-tracking data were used to study participants' spontaneous gaze-dynamics in the VR classroom throughout the entire trial and also to examine whether background Sound-events elicited overt gaze-shifts. Gaze-data preprocessing involved removing measurements around blinks (from −100 to 200 ms after) and any other data points marked as 'missing data' by the eye-tracker. Gaze-data were analyzed using Python, closely following recommended procedures for analysis of gaze-data in VR (*Anderson et al., 2023*). To assess gaze-patterns and detect gaze-shift, we calculated the Euclidean distance between adjacent data points, using the *x*, *y*, *z* coordinates in 3D space provided by the eye-tracker. 'Fixations' were defined as clusters of adjacent time-points within a radius of less than 0.01 distance from each other, lasting at least 80 ms. Each fixation was associated with a specific ROI in the VR classroom, indicating the object in the virtual space that the participant looked at. 'Gaze-shifts' were defined as transitions between two stable fixations.

To analyze spontaneous gaze-dynamics throughout an entire trial, we quantified the percent of each trial that participants spent looking at the teacher (target) versus all other ROIs, as well as the number of gaze-shifts away from the teacher per trial. Between-group differences in these measures of spontaneous gaze-patterns were assessed using unpaired *t*-tests, separately for the Quiet and Events conditions. The $BF_{10}$ was calculated for each pairwise comparison to test the level of confidence for embracing or rejecting the null hypothesis.

To determine whether background Sound-events elicited overt gaze-shifts, we segmented the gaze-data into 2-s long epochs around each Event-sound. As a control, we randomly chose a similar number of 2-s long epochs from the Quiet-condition. We then counted how many epochs included at least one gaze-shift away from the teacher, and used a paired *t*-test to determine whether the frequency of gaze-shifts was higher following Sound-events relative to the Control epochs (regardless of Group). We also used a mixed ANOVA to test for potential differences in the likelihood of gaze-shifts following the different event-types (Artificial vs. Human non-verbal), and for the two Groups (AD(H)D vs. Control). For this analysis, the proportion of gaze-shifts was normalized for each participant by subtracting the number of gaze-shifts following Sound-events relative to the control epochs.

## Multivariate analyses

Besides testing for effects of Group in each metric individually, we also performed multivariate analyses to examine the relationship between the various metrics collected here and their ability to predict ADHD diagnosis (binary) and/or the severity of AD(H)D symptoms.

This analysis focused on 10 key measures, based on the results of the univariate analyses (averaged across conditions, when relevant): 5 measures related to the response to the teacher's speech: behavioral performance, neural speech tracking decoding (reconstruction correlation), the N1- and P2 peaks of the speech tracking response (TRF-N1; TRF-P2), and the average number of gaze-shifts away from the teacher; 3 measures related to responses to the sound-events: amplitude of the N1- and P2-responses to sound-events (ERP-N1; ERP-P2), phasic SC response to sound-events (event-related SC); and 2 global measures associated with the listeners' 'state': alpha- and beta-power.

First, we calculated Spearman's correlation between all pairs of measures (fdr correction for multiple comparisons was applied). Next, we entered all factors into a multiple regression model and performed a dominance analysis to determine the relative importance of each measure for predicting whether an individual was in the ADHD or control group (logistic regression) or the severity of ADHD symptoms, as reflected by their ASRS scores (linear regression). Dominance analysis was performed as follows: For each of the 10 measures, we computed the difference in a model-fit index (AUC for logistic regression; $R^2$ for linear regression) between a full model containing all 10 predictors, and a model where the measure under consideration was held-out, as a means for assessing the unique contribution of each measure to overall model performance (dominance). We then sorted the measures based on their dominance and report the multiple regression model that explains the most

amount of variance using the least number of variables. These analyses were conducted using the statistical software JASP (*Team, 2024*).

## Acknowledgements

This work was supported by the Israel Science Foundation (ISF grant # 2339/20 to EZG), the National Institute of Mental Health (NIMH grant R33MH110043 to JS), and the Binational Science Foundation (BSF grant # 2022024 to EZG and JS). We would also like to thank the Minerva Center for Human Intelligence in Immersive, Augmented and Mixed Realities at Tel Aviv University, for supporting for this research.

## Additional information

### Funding

| Funder | Grant reference number | Author |
| --- | --- | --- |
| United States - Israel Binational Science Foundation | 2022024 | Julie B Schweitzer<br>Elana Zion Golumbic |
| Israel Science Foundation | 2339/20 | Elana Zion Golumbic |
| National Institute of Mental Health | MH135266-02 | Julie B Schweitzer<br>Elana Zion Golumbic |
| National Institute of Mental Health | R33MH110043 | Julie B Schweitzer |

The funders had no role in study design, data collection, and interpretation, or the decision to submit the work for publication.

### Author contributions

Orel Levy, Data curation, Formal analysis, Investigation, Visualization, Methodology, Writing – original draft, Writing – review and editing; Shirley Libman Hackmon, Conceptualization, Data curation, Formal analysis; Yair Zvilichovsky, Conceptualization, Software, Methodology; Adi Korisky, Supervision, Methodology, Project administration; Aurelie Bidet-Caulet, Conceptualization, Writing – review and editing; Julie B Schweitzer, Conceptualization, Resources, Writing – review and editing; Elana Zion Golumbic, Conceptualization, Resources, Data curation, Formal analysis, Supervision, Funding acquisition, Investigation, Visualization, Methodology, Writing – original draft, Project administration, Writing – review and editing

### Author ORCIDs

Orel Levy ![ORCID] https://orcid.org/0009-0008-9075-2818
Aurelie Bidet-Caulet ![ORCID] http://orcid.org/0009-0002-8135-0725
Elana Zion Golumbic ![ORCID] https://orcid.org/0000-0002-8831-3188

### Ethics

The experimental protocol was approved by the ethics committee at Bar-Ilan University (protocol # ISU202112002), and all participants provided written informed consent prior to their involvement in the study and data collection procedures.

Reviewer #1 (Public review): https://doi.org/10.7554/eLife.103235.3.sa1
Reviewer #2 (Public review): https://doi.org/10.7554/eLife.103235.3.sa2
Reviewer #3 (Public review): https://doi.org/10.7554/eLife.103235.3.sa3
Author response https://doi.org/10.7554/eLife.103235.3.sa4

## Additional files

### Supplementary files
MDAR checklist

### Data availability
Video examples of the VR classroom and the entire dataset are publicly available on OSF: https://osf.io/svjqg. Please contact authors for additional information on data structure.

The following dataset was generated:

| Author(s) | Year | Dataset title | Dataset URL | Database and Identifier |
|---|---|---|---|---|
| Levy O, Zion Golumbic E | 2025 | Selective attention and sensitivity to auditory disturbances in a virtually real Classroom | https://osf.io/svjqg | Open Science Framework, svjqg |

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
