## [Editor Report · eLife Assessment]

This **important** study investigates how AD(H)D affects attention using neural and physiological measures in a Virtual Reality (VR) environment. **Solid** evidence is provided that individuals diagnosed with AD(H)D differ from control participants in both the encoding of the target sound and the encoding of acoustic interference. The VR paradigm here can potentially bridge lab experiments and real-life experiments. The study is of potential interests to researchers who are interested in auditory cognition, education, and ADHD.

---

## [Referee Report · Reviewer #1 (Public review)]

Summary:

This is an interesting study on AD(H)D. The authors combine a variety of neural and physiological metrics to study attention in a VR classroom setting. The manuscript is well written and the results are interesting, ranging from an effect of group (AD(H)D vs. control) on metrics such as envelope tracking, to multivariate regression analyses considering alpha-power, gaze, TRF, ERPs, and behaviour simultaneously. I find the first part of the results clear and strong. The multivariate analyses in Tables 1 and 2 are good ideas, but I think they would benefit from additional clarifications. Overall, I think that the methodological approach is useful in itself. The rest is interesting in that it informs us on which metrics are sensitive to group-effects and correlated with each other. I think this might be one interesting way forward. Indeed, much more work is needed to clarify how these results change with different stimuli and tasks. So, I see this as an interesting first step into more naturalistic measurement of speech attention.

Strengths:

I praise the authors for this interesting attempt to tackle a challenging topic with naturalistic experiment and metrics. I think the results broadly make sense and they contribute to a complex literature that is far from being linear and cohesive.

Weaknesses:

The authors have successfully addressed most of my concerns during the review process. Some weaknesses remain in this resubmission, but they do not make the results invalid. For example:

- The EEG data was filtered twice, which is not recommended as that can introduce additional filtering artifacts. So, while I definitely do not recommend doing that, I do not expect that issue to have an impact on this specific result.

- The authors did not check whether participants were somewhat familiar with the topics in the speech material. The authors agreed that this point might be beneficial for future research.

- The hyperparameter tuning is consistent with previous work from the authors, and it involves selecting the optimal lambda value of the regularized regression based on the group average, thus choosing a single lambda value for all participants. In my opinion, that is not the optimal way to run those models, and I do not generally recommend using this approach. The reason is that the lambda can change depending on the magnitude of the signals and the SNR, leading to different optimal lambdas for distinct participants. On the other hand, finding those optimal lambda values for individual participants can be difficult depending on the amount of data and amount of noise, so it is sometimes necessary to apply strategies that ensure an appropriate choice of lambda. Using the group average as a metric for hyperparameter tuning produces a more stable metric and lambda value selection, which might be preferrable (even though this choice should not be taken lightly). In this specific case, I think the authors had a good reason to do so.

Comments on revisions:

The authors have done a great job at addressing my comments. I don't have any further concerns. Congratulations!

---

## [Referee Report · Reviewer #2 (Public review)]

Summary:

While selective attention is a crucial ability of human beings, previous studies on selective attention are primarily conducted in a strictly controlled context, leaving a notable gap in underlying the complexity and dynamic nature of selective attention in a naturalistic context. This issue is particularly important for classroom learning in individuals with ADHD, as selecting the target and ignoring the distractions are pretty difficult for them but are the pre-requirement of effective learning. The authors of this study have addressed this challenge using a well-motivated study. I believe the findings of this study will be a nice addition to the fields both cognitive neuroscience and educational neuroscience.

Strengths:

To achieve the purpose of setting up a naturalistic context, the authors have based their study on a novel Virtual Reality platform. This is clever as it is usually difficult to perform such a study in the real classroom. Moreover, various techniques such as brain imaging, eye-tracking and physiological measurement are combined to collect multi-level data. They found that, different from the controls, individuals with ADHD had higher neural responses to the irrelevant rather than the target sounds, reduced speech tracking of the teacher. Additionally, the power of alpha-oscillations and frequency of gaze-shifts away from the teacher are found to be associated with the ADHD symptoms. These results provide new insights into the mechanism of selective attention among ADHD populations.

Weaknesses:

It is worth noting that nowadays there has been some studies trying to do so in the real classroom, and thus the authors should acknowledge the difference between the virtual and real classroom context and foresee the potential future changes.

The approach of combining multi-level data owns advantage to obtain reliable results, but also raises significant difficult for the readers to understand the main results.

- An appraisal of whether the authors achieved their aims, and whether the results support their conclusions.

As expected, individuals with ADHD showed anomalous pattern of neural responses, and eye-tracking pattern, compared to the controls. But there are also some similarities between groups such as amount of time paying attention to teachers, etc. In general, their conclusions are supported.

- A discussion of the likely impact of the work on the field, and the utility of the methods and data to the community.

The findings are an extension of previous efforts in understanding selective attention in the naturalistic context. The findings of this study are particularly helpful in inspiring teacher's practice and advancing the research of educational neuroscience. This study demonstrates, again, that it is important to understand the complexity of cognitive process in the naturalistic context.

Comments on revisions:

The authors have appropriately responded to my concerns. I do not have other comments. I do hope to see more data and results from the authors in future.

---

## [Referee Report · Reviewer #3 (Public review)]

Summary:

The authors conducted a well-designed experiment, incorporating VR classroom scenes and background sound events, with both control and ADHD participants. They employed multiple neurophysiological measures, such as EEG, eye movements, and skin conductance, to investigate the mechanistic underpinnings of paying attention in class and the disruptive effects of background noise.

The results revealed that individuals with ADHD exhibited heightened sensory responses to irrelevant sounds and reduced tracking of the teacher's speech. Overall, this manuscript presented an ecologically valid paradigm for assessing neurophysiological responses in both control and ADHD groups. The analyses were comprehensive and clear, making the study potentially valuable for the application of detecting attentional deficits.

Strengths:

• The VR learning paradigm is well-designed and ecologically valid.

• The neurophysiological metrics and analyses are comprehensive, and two physiological markers are identified capable of diagnosing ADHD.

• The data shared could serve as a benchmark for future studies on attention deficits in ecologically valid scenarios.

Weaknesses:

• Several results are null results, i.e., no significant differences were found between ADHD and control populations.

Comments on revisions:

The authors have addressed all of my concerns with the original manuscript.

---

## [Author Response]

The following is the authors’ response to the original reviews

**Reviewer #1:**
(1) Line numbers are missing.

Added

(2) VR classroom. Was this a completely custom design based on Unity, or was this developed on top of some pre-existing code? Many aspects of the VR classroom scenario are only introduced (e.g., how was the lip-speech synchronisation done exactly?). Additional detail is required. Also, is or will the experiment code be shared publicly with appropriate documentation? It would also be useful to share brief example video-clips.

We have added details about the VR classroom programming to the methods section (p. 6-7), and we have now included a video-example as supplementary material.

“Development and programming of the VR classroom were done primarily in-house, using assets (avatars and environment) were sourced from pre-existing databases. The classroom environment was adapted from assets provided by Tirgames on TurboSquid (https://www.turbosquid.com/Search/Artists/Tirgames) and modified to meet the experimental needs. The avatars and their basic animations were sourced from the Mixamo library, which at the time of development supported legacy avatars with facial blendshapes (this functionality is no longer available in current versions of Mixamo). A brief video example of the VR classroom is available at: https://osf.io/rf6t8.

“To achieve realistic lip-speech synchronization, the teacher’s lip movements were controlled by the temporal envelope of the speech, adjusting both timing and mouth size dynamically. His body motions were animated using natural talking gestures.”

While we do intent to make the dataset publicly available for other researchers, at this point we are not making the code for the VR classroom public. However, we are happy to share it on an individual-basis with other researchers who might find it useful for their own research in the future.

(3) "normalized to the same loudness level using the software Audacity". Please specify the Audacity function and parameters.

We have added these details (p.7)

“All sound-events were normalized to the same loudness level using the Normalize function in the audio-editing software Audacity (theaudacityteam.org, ver 3.4), with the peak amplitude parameter set to -5 dB, and trimmed to a duration of 300 milliseconds.“

(4) Did the authors check if the participants were already familiar with some of the content in the mini-lectures?

This is a good point. Since the mini-lectures spanned many different topics, we did not pre-screen participants for familiarity with the topics, and it is possible that some of the participants had some pre-existing knowledge.

In hindsight, it would have been good to have added some reflective questions regarding participants prior knowledge as well as other questions such as level of interest in the topic and/or how well they understood the content. These are elements that we hope to include in future versions of the VR classroom.

(5) "Independent Component Analysis (ICA) was then used to further remove components associated with horizontal or vertical eye movements and heartbeats". Please specify how this selection was carried out.

Selection of ICA components was done manually based on visual inspection of their time-course patterns and topographical distributions, to identify components characteristic of blinks, horizontal eye-movements and heartbeats. Examples of these distinct components are provided in Author response image 1 below. These is now specified in the methods section.

**Author response image 1. sa4fig1:** 

(6) "EEG data was further bandpass filtered between 0.8 and 20 Hz". If I understand correctly, the data was filtered a second time. If that's the case, please do not do that, as that will introduce additional and unnecessary filtering artifacts. Instead, the authors should replace the original filter with this one (so, filtering the data only once). Please see de Cheveigne and Nelkn, Neuron, 2019 for an explanation. Also, please provide an explanation of the rationale for further restricting the cut-off bands in the methods section. Finally, further details on the filters should be included (filter type and order, for example).

Yes, the data was indeed filtered twice. The first filter is done as part of the preprocessing procedure, in order to remove extremely high- and low- frequency noise but retain most activity within the range of “neural” activity. This broad range is mostly important for the ICA procedure, so as to adequately separate between ocular and neural contribution to the recorded signal.

However, since both the speech tracking responses and ERPs are typically less broadband and are comprised mostly of lower frequencies (e.g., those that make up the speech-envelope), a second narrower filter was applied to improve TRF model-fit and make ERPs more interpretable.

In both cases we used a fourth order zero-phase Butterworth IIR filter with 1-seconds of padding, as implemented in the Fieldtrip toolbox. We have added these details to the manuscript.

(7) "(~ 5 minutes of data in total), which is insufficient for deriving reliable TRFs". That is a bit pessimistic and vague. What does "reliable" mean? I would tend to agree when talking about individual subject TRFs, which 5 min per participant can be enough at the group level. Also, this depends on the specific speech material. If the features are univariate or multivariate. Etc. Please narrow down and clarify this statement.

We determined that the data in the Quiet condition (~5 min) was insufficient for performing reliable TRF analysis, by assessing whether its predictive-power was significantly better than chance. As shown in Author response image 2 below, the predictive power achieved using this data was not higher than values obtained in permuted data (p = 0.43). Therefore, we did not feel that it was appropriate to include TRF analysis of the Quiet condition in this manuscript. We have now clarified this in the manuscript (p. 10)

**Author response image 2. sa4fig2:** 

(8) "Based on previous research in by our group (Kaufman & Zion Golumbic 2023), we chose to use a constant regularization ridge parameter (λ = 100) for all participants and conditions". This is an insufficient explanation. I understand that there is a previous paper involved. However, such an unconventional choice that goes against the original definition and typical use of these methods should be clearly reported in this manuscript.

We apologize for not clarifying this point sufficiently, and have added an explanation of this methodological choice (p.11):

“The mTRF toolbox uses a ridge-regression approach for L2 regularization of the model to ensure better generalization to new data. We tested a range of ridge parameter values (λ's) and used a leave-one-out cross-validation procedure to assess the model’s predictive power, whereby in each iteration, all but one trials are used to train the model, and it is then applied to the left-out trial. The predictive power of the model (for each λ) is estimated as the Pearson’s correlation between the predicted neural responses and the actual neural responses, separately for each electrode, averages across all iterations. We report results of the model with the λ the yielded the highest predictive power at the group-level (rather than selecting a different λ for each participant which can lead to incomparable TRF models across participants; see discussion in Kaufman & Zion Golumbic 2023).”

Assuming that the explanation will be sufficiently convincing, which is not a trivial case to make, the next issue that I will bring up is that the lambda value depends on the magnitude of input and output vectors. While the input features are normalised, I don't see that described for the EEG signals. So I assume they are not normalized. In that case, the lambda would have at least to be adapted between subjects to account for their different magnitude.

We apologize for omitting this detail – yes, the EEG signals were normalized prior to conducting the TRF analysis. We have updated the methods section to explicitly state this pre-processing step (p.10).

Another clarification, is that value (i.e., 100) would not be comparable either across subjects or across studies. But maybe the authors have a simple explanation for that choice? (note that this point is very important as this could lead others to use TRF methods in an inappropriate way - but I understand that the authors might have specific reasons to do so here). Note that, if the issue is finding a reliable lambda per subject, a more reasonable choice would be to use a fixed lambda selected on a generic (i.e., group-level) model. However selecting an arbitrary lambda could be problematic (e.g., would the results replicate with another lambda; and similarly, what if a different EEG system was used, with different overall magnitude, hence the different impact of the regularisation).

We fully agree that selecting an arbitrary lambda is problematic (esp across studies). As clarified above, the group-level lambda chosen here for the encoding more was data-driven, optimized based on group-level predictive power.

(9) "L2 regularization of the model, to reduce its complexity". Could the authors explain what "reduce its complexity" refers to?

Our intension here was to state that the L2 regularization constrains the model’s weights so that it can better generalize between to left-out data. However, for clarity we have now removed this statement.

(10) The same lambda value was used for the decoding model. From personal experience, that is very unlikely to be the optimal selection. Decoding models typically require a different (usually larger) lambda than forward models, which can be due to different reasons (different SNR of "input" of the model and, crucially, very different dimensionality).

We agree with the reviewer that treatment of regularization parameters might not be identical for encoding and decoding models. Our initial search of lambda parameters was limited to λ = 0.01 - 100, with λ = 100 showing the best reconstruction correlations. However, following the reviewer’s suggestion we have now broadened the range and found that, in fact reconstruction correlations are further improved and the best lambda is λ = 1000 (see Author response image 3 below, left panel). Importantly, the difference in decoding reconstruction correlations between the groups is maintained regardless of the choice of lambda (although the effect-size varies; see Author response image 3, right panel). We have now updated the text to reflect results of the model with λ = 1000.

**Author response image 3. sa4fig3:** 

(11) Skin conductance analysis. Additional details are required. For example, how was the linear interpolation done exactly? The raw data was downsampled, sure. But was an anti-aliasing filter applied? What filter exactly? What implementation for the CDA was run exactly?

We have added the following details to the methods section (p. 14):

“The Skin Conductance (SC) signal was analyzed using the Ledalab MATLAB toolbox (version 3.4.9; Benedek and Kaernbach, 2010; http://www.ledalab.de/) and custom-written scripts. The raw data was downsampled to 16Hz using FieldTrip's ft_resampledata function, which applies a built-in anti-aliasing low-pass filter to prevent aliasing artifacts. Data were inspected manually for any noticeable artifacts (large ‘jumps’), and if present were corrected using linear interpolation in Ledalab. A continuous decomposition analysis (CDA) was employed to separate the tonic and phasic SC responses for each participant. The CDA was conducted using the 'sdeco' mode (signal decomposition), which iteratively optimizes the separation of tonic and phasic components using the default regularization settings.”

(12) "N1- and P2 peaks of the speech tracking response". Have the authors considered using the N1-P2 complex rather than the two peaks separately? Just a thought.

This is an interesting suggestion, and we know that this has been used sometimes in more traditional ERP literature. In this case, since neither peak was modulated across groups, we did not think this would yield different results. However, it is a good point to keep in mind for future work.

(13) Figure 4B. The ticks are missing. From what I can see (but it's hard without the ticks), the N1 seems later than in other speech-EEG tracking experiments (where is closer to ~80ms). Could the authors comment on that? Or maybe this looks similar to some of the authors' previous work?

We apologize for this and have added ticks to the figure.

In terms of time-course, a N1 peak at around 100ms is compatible with many of our previous studies, as well as those from other groups.

(14) Figure 4C. Strange thin vertical grey bar to remove.

Fixed.

(15) Figure 4B: What about the topographies for the TRF weights? Could the authors show that for the main components?

Yes. The topographies of the main TRF components are similar to those of the predictive power and are compatible with auditory responses. We have added them to Figure 4B.

(16) Figure 4B: I just noticed that this is a grand average TRF. That is ok (but not ideal) only because the referencing is to the mastoids. The more appropriate way of doing this is to look at the GFP, instead, which estimates the presence of dipoles. And then look at topographies of the components. Averaging across channels makes the plotted TRF weaker and noisier. I suggest adding the GFP to the plot. Also, the colour scale in Figure 4A is deceiving, as blue is usually used for +/- in plots of the weights. While that is a heatmap, where using a single colour or even yellow to red would be less deceiving at first look. Only cosmetics, indeed. The result is interesting nonetheless!

We apologize for this, and agree with the reviewer that it is better not to average across EEG channels. In the revised Figure, we now show the TRFs based on the average of electrodes FC1, FC2, and FCz, which exhibited the strongest activity for the two main components.

Following the previous comment, we have also included the topographical representation of the TRF main components, to give readers a whole-head perspective of the TRF.

We have also fixed the color-scales.

We are glad that the reviewer finds this result interesting!

(17) Figure 4C. This looks like a missed opportunity. That metric shows a significant difference overall. But is that underpinned but a generally lower envelope reconstruction correlation, or by a larger deviation in those correlations (so, that metric is as for the control in some moments, but it drops more frequently due to distractibility)?

We understand the reviewer’s point here, and ideally would like to be able to address this in a more fine-grained analysis, for example on a trial-by-trial basis. However, the design of the current experiment was not optimized for this, in terms of (for example) number of trials, the distribution of sound-events and behavioral outcomes. We hope to be able to address this issue in our future research.

(18) I am not a fan of the term "accuracy" for indicating envelope reconstruction correlations. Accuracy is a term typically associated with classification. Regression models are typically measured through errors, loss, and sometimes correlations. 'Accuracy' is inaccurate (no joke intended).

We accept this comment and now used the term “reconstruction correlation”.

(19) Discussion. "The most robust finding in". I suggest using more precise terminology. For example, "largest effect-size".

We agree and have changed the terminology (p. 31).

(20) "individuals who exhibited higher alpha-power [...]". I probably missed this. But could the authors clarify this result? From what I can see, alpha did not show an effect on the group. Is this referring to Table 2? Could the authors elaborate on that? How does that reconcile with the non-significant effect of the group? In that same sentence, do you mean "and were more likely"? If that's the case, and they were more likely to report attentional difficulties, how is it that there is no group-effect when studying alpha?

Yes, this sentence refers to the linear regression models described in Figure 10 and in Table 2. As the reviewer correctly points out, this is one place where there is a discrepancy between the results of the between-group analysis (ADHD diagnosis yes/no) and the regression analysis, which treats ADHD symptoms as a continuum, across both groups. The same is true for the gaze-shift data, which also did not show a significance between-group effect but was identified in the regression analysis as contributing to explaining the variance in ADHD symptoms.

We discuss this point on pages 30-31, noting that “although the two groups are clearly separable from each other, they are far from uniform in the severity of symptoms experienced”, which motivated the inclusion of both analyses in this paper.

At the bottom of p. 31 we specifically address the similarities and differences between the between-group and regression-based results. In our opinion, this pattern emphasizes that while neither approach is ‘conclusive’, looking at the data through both lenses contributes to an overall better understanding of the contributing factors, as well as highlighting that “no single neurophysiological measure alone is sufficient for explaining differences between the individuals – whether through the lens of clinical diagnosis or through report of symptoms”.

(21) "why in the latter case the neural speech-decoding accuracy did not contribute to explaining ASRS scores [...]". My previous point 1 on separating overall envelope decoding from its deviation could help there. The envelope decoding correlation might go up and down due to SNR, while you might be more interested in the dynamics over time (i.e., looking at the reconstructions over time).

Again, we appreciate this comment, but believe that this additional analysis is outside the scope of what would be reliably-feasible with the current dataset. However, since the data will be made publicly available, perhaps other researchers will have better ideas as to how to do this.

(22) Data and code sharing should be discussed. Also, specific links/names and version numbers should be included for the various libraries used.

We are currently working on organizing the data to make it publicly available on the Open Science Project.

We have updated links and version numbers for the various toolboxes/software used, throughout the manuscript.

**Reviewer #2:**
(1) While it is highly appreciated to study selective attention in a naturalistic context, the readers would expect to see whether there are any potential similarities or differences in the cognitive and neural mechanisms between contexts. Whether the classic findings about selective attention would be challenged, rebutted, or confirmed? Whether we should expect any novel findings in such a novel context? Moreover, there are some studies on selective attention in the naturalistic context though not in the classroom, it would be better to formulate specific hypotheses based on previous findings both in the strictly controlled and naturalistic contexts.

Yes, we fully agree that comparing results across different contexts would be extremely beneficial and important.

The current paper serves as an important proof-first-concept demonstrating the plausibility and scientific potential of using combined EEG-VR-eyetracking to study neurophysiological aspects of attention and distractibility, but is also the basis for formulating specific hypothesis that will be tested in follow-up studies.

If fact, a follow up study is already ongoing in our lab, where we are looking into this point, by testing users in different VR scenarios (e.g., classroom, café, office etc.), and assessing whether similar neurophysiological patterns are observed across contexts and to what degree they are replicable within and across individuals. We hope to share these data with the community in the near future.

(2) Previous studies suggest handedness and hemispheric dominance might impact the processing of information in each hemisphere. Whether these issues have been taken into consideration and appropriately addressed?

This is an interesting point. In this study we did not specifically control for handedness/hemispheric dominance, since most of the neurophysiological measured used here are sensory/auditory in their nature, and therefore potentially invariant to handedness. Moreover, the EEG signal is typically not very sensitive to hemispheric dominance, at least for the measures used here. However, this might be something to consider more explicitly in future studies. Nonetheless, we have added handedness information to the Methods section (p. 5): “46 right-handed, 3 left-handed”

(3) It would be interesting to know how students felt about the Virtual Classroom context, whether it is indeed close to the real classroom or to some extent different.

Yes, we agree. Obviously, the VR classroom differs in many ways from a real classroom, in terms of the perceptual experience, social aspects and interactive possibilities. We did ask participants about their VR experience after the experiment, and most reported feeling highly immersed in the VR environment and engaged in the task, with a strong sense of presence in the virtual-classroom.

We note that, in parallel to the VR studies in our lab, we are also conducting experiments in real classrooms, and we hope that the cross-study comparison will be able to shed more light on these similarities/differences.

(4) One intriguing issue is whether neural tracking of the teacher's speech can index students' attention, as the tracking of speech may be relevant to various factors such as sound processing without semantic access.

Another excellent point. While separating the ‘acoustic’ and ‘semantic’ contributions to the speech tracking response is non-trivial, we are currently working on methodological approaches to do this (again, in future studies) following, for example, the hierarchical TRF approach used by Brodbeck et al. and others.

(5) There are many results associated with various metrics, and many results did not show a significant difference between the ADHD group and the control group. It is difficult to find the crucial information that supports the conclusion. I suggest the authors reorganize the results section and report the significant results first, and to which comparison(s) the readers should pay attention.

We apologize if the organization of the results section was difficult to follow. This is indeed a challenge when collecting so many different neurophysiological metrics.

To facilitate this, we have now added a paragraph at the beginning of the result section, clarifying its structure (p.16):

The current dataset is extremely rich, consisting of many different behavioral, neural and physiological responses. In reporting these results, we have separated between metrics that are associated with paying attention to the teacher (behavioral performance, neural tracking of the teacher’s speech, and looking at the teacher), those capturing responses to the irrelevant sound-events (ERPs and event-related changes in SC and gaze); as well as more global neurophysiological measures that may be associated with the listeners’ overall ‘state’ of attention or arousal (alpha- and beta-power and tonic SC).

Moreover, within each section we have ordered the analysis such that the ones with significant effects are first. We hope that this contributes to the clarity of the results section.

(6) The difference between artificial and non-verbal humans should be introduced earlier in the introduction and let the readers know what should be expected and why.

We have added this to the Introduction (p. 4)

(7) It would be better to discuss the results against a theoretical background rather than majorly focusing on technical aspects.

We appreciate this comment. In our opinion, the discussion does contain a substantial theoretical component, both regarding theories of attention and attention-deficits, and also regarding their potential neural correlates. However, we agree that there is always room for more in depth discussion.

**Reviewer #3:**
Major:(1) While the study introduced a well-designed experiment with comprehensive physiological measures and thorough analyses, the key insights derived from the experiment are unclear. For example, does the high ecological validity provide a more sensitive biomarker or a new physiological measure of attention deficit compared to previous studies? Or does the study shed light on new mechanisms of attention deficit, such as the simultaneous presence of inattention and distraction (as mentioned in the Conclusion)? The authors should clearly articulate their contributions.

Thanks for this comment.

We would not say that this paper is able to provide a ‘more sensitive biomarker’ or a ‘new physiological measure of attention’ – in order to make those type of grand statements we would need to have much more converging evidence from multiple studies and using both replication and generalization approaches.

Rather, from our perspective, the key contribution of this work is in broadening the scope of research regarding the neurophysiological mechanisms involved in attention and distraction.

Specifically, this work:

(1) Offers a significant methodological advancement of the field – demonstrating the plausibility and scientific potential of using combined EEG-VR-eyetracking to study neurophysiological aspects of attention and distractibility in contexts that ‘mimic’ real-life situations (rather than highly controlled computerized tasks).

(2) Provides a solid basis formulating specific mechanistic hypothesis regarding the neurophysiological metrics associated with attention and distraction, the interplay between them, and their potential relation to ADHD-symptoms. Rather than being an end-point, we see these results as a start-point for future studies that emphasize ecological validity and generalizability across contexts, that will hopefully lead to improved mechanisms understanding and potential biomarkers of real-life attentional capabilities (see also response to Rev #2 comment #1 above).

(3) Highlights differences and similarities between the current results and those obtained in traditional ‘highly controlled’ studies of attention (e.g., in the way ERPs to sound-events differ between ADHD and controls; variability in gaze and alpha-power; and more broadly about whether ADHD symptoms do or don’t map onto specific neurophysiological metrics). Again, we do not claim to give a definitive ’answer’ to these issues, but rather to provide a new type of data that can expands the conversation and address the ecological validity gap in attention research.

(2) Based on the multivariate analyses, ASRS scores correlate better with the physiological measures rather than the binary deficit category. It may be worthwhile to report the correlation between physiological measures and ASRS scores for the univariate analyses. Additionally, the correlation between physiological measures and behavioral accuracy might also be interesting.

Thanks for this. The beta-values reported for the regression analysis reflect the correlations between the different physiological measures and the ASRS scores (p. 30). From a statistical perspective, it is better to report these values rather than the univariate correlation-coefficients, since these represent the ‘unique’ relationship with each factor, after controlling for all the others.

The univariate correlations between the physiological measures themselves, as well as with behavioral accuracy, are reported in Figure 10

(3) For the TRF and decoding analysis, the authors used a constant regularization parameter per a previous study. However, the optimal regularization parameter is data-dependent and may differ between encoding and decoding analyses. Furthermore, the authors did not conduct TRF analysis for the quiet condition due to the limited ~5 minutes of data. However, such a data duration is generally sufficient to derive a stable TRF with significant predictive power (74).

The reviewer raises two important points, also raised by Rev #1 (see above).

Regarding the choice of regularization parameters, we have now clarified that although we used a common lambda value for all participants, it was selected in a data-driven manner, so as to achieve an optimal predictive power at the group-level.

See revised methods section:

“The mTRF toolbox uses a ridge-regression approach for L2 regularization of the model to ensure better generalization to new data. We tested a range of ridge parameter values (λ's) and used a leave-one-out cross-validation procedure to assess the model’s predictive power, whereby in each iteration, all but one trials are used to train the model, and it is then applied to the left-out trial. The predictive power of the model (for each λ) is estimated as the Pearson’s correlation between the predicted neural responses and the actual neural responses, separately for each electrode, averages across all iterations. We report results of the model with the λ the yielded the highest predictive power at the group-level (rather than selecting a different λ for each participant which can lead to incomparable TRF models across participants; see discussion in Kaufman & Zion Golumbic 2023).”

Regarding whether data was sufficient in the Quiet condition for performing TRF analysis – we are aware of the important work by Mesik & Wojtczak, and had initially used this estimate when designing our study. However, when assessing the predictive-power of the TRF model trained on data from the Quiet condition, we found that it was not significantly better than chance (see Author response image 2, ‘real’ predictive power vs. permuted data). Therefore, we ultimately did not feel that it was appropriate to include TRF analysis of the Quiet condition in this manuscript. We have now clarified this in the manuscript (p. 10)

(4) As shown in Figure 4, for ADHD participants, decoding accuracy appears to be lower than the predictive power of TRF. This result is surprising because more data (i.e., data from all electrodes) is used in the decoding analysis.

This is an interesting point – however, in our experience it is not necessarily the case that decoding accuracy (i.e., reconstruction correlation with the stimulus) is higher than encoding predictive-power. While both metrics use Pearson’s’ correlations, they quantify the similarity between two different types of signals (the EEG and the speech-envelope). Although the decoding procedure does use data from all electrodes, many of them don’t actually contain meaningful information regarding the stimulus, and thus could just as well hinder the overall performance of the decoding.

(5) Beyond the current analyses, the authors may consider analyzing inter-subject correlation, especially for the gaze signal analysis. Given that the area of interest during the lesson changes dynamically, the teacher might not always be the focal point. Therefore, the correlation of gaze locations between subjects might be better than the percentage of gaze duration on the teacher.

Thanks for this suggestion. We have tried to look into this, however working with eye-gaze in a 3-D space is extremely complex and we are not able to calculate reliable correlations between participants.

(6) Some preprocessing steps relied on visual and subjective inspection. For instance, " Visual inspection was performed to identify and remove gross artifacts (excluding eye movements) " (P9); " The raw data was downsampled to 16Hz and inspected for any noticeable artifacts " (P13). Please consider using objective processes or provide standards for subjective inspections.

We are aware of the possible differences between objective methods of artifact rejection vs. use of manual visual inspection, however we still prefer the manual (subjective) approach. As noted, in this case only very large artifacts were removed, exceeding ~ 4 SD of the amplitude variability, so as to preserve as many full-length trials as possible.

(7) Numerous significance testing methods were employed in the manuscript. While I appreciate the detailed information provided, describing these methods in a separate section within the Methods would be more general and clearer. Additionally, the authors may consider using a linear mixed-effects model, which is more widely adopted in current neuroscience studies and can account for random subject effects.

Indeed, there are many statistical tests in the paper, given the diverse types of neurophysiological data collected here. We actually thought that describing the statistics per method rather than in a separate “general” section would be easier to follow, but we understand that readers might diverge in their preferences.

Regarding the use of mixed-effect models – this is indeed a great approach. However, it requires deriving reliable metrics on a per-trial basis, and while this might be plausible for some of our metrics, the EEG and GSR metrics are less reliable at this level. This is why we ultimately chose to aggregate across trials and use a regular regression model rather than mixed-effects.

(8) Some participant information is missing, such as their academic majors. Given that only two lesson topics were used, the participants' majors may be a relevant factor.

To clarify – the mini-lectures presented here actually covered a large variety of topics, broadly falling within the domains of history, science and social-science and technology. Regarding participants’ academic majors, these were relatively diverse, as can be seen in Author response table 1 and Author response image 4.

**Author response table 1. sa4table1:** 

Academic major	Count
Not a student	14
Neuroscience	7
Middle Esstern Studies	2
Criminology	2
Psychology and Economics	1
Business Administration	1
Veterinary Medicine	1
Computer Engineering	1
Arabic and Middle Eastern Studies	1
Computer Science	1
Psychology and Special Education	1
Agriculture	1
Communication and Political Science	1
Materials Engineering	1
Medicine	1
Criminology and Special Education	1
Music	1
Arabic	1
Nursing	1
Education	1
Psychology	1
Electrical Engineering	1
Psychology and History	1
Information Systerns	1
Sociology and Anthropology	1
Jewish Art	1
Law	1
Life Sciences	1

**Author response image 4. sa4fig4:** 

(9) Did the multiple regression model include cross-validation? Please provide details regarding this.

Yes, we used a leave-one-out cross validation procedure. We have now clarified this in the methods section which now reads:

“The mTRF toolbox uses a ridge-regression approach for L2 regularization of the model to ensure better generalization to new data. We tested a range of ridge parameter values (λ's) and used a leave-one-out cross-validation procedure to assess the model’s predictive power, whereby in each iteration, all but one trials are used to train the model, and it is then applied to the left-out trial. The predictive power of the model (for each λ) is estimated as the Pearson’s correlation between the predicted neural responses and the actual neural responses, separately for each electrode, averages across all iterations. We report results of the model with the λ the yielded the highest predictive power at the group-level (rather than selecting a different λ for each participant which can lead to incomparable TRF models across participants; see discussion in Kaufman & Zion Golumbic 2023).”

Minor:(10) Typographical errors: P5, "forty-nine 49 participants"; P21, "$ref"; P26, "Table X"; P4, please provide the full name for "SC" when first mentioned.

Thanks! corrected